# Self-Supervised Representation Learning on Neural Network Weights for Model Characteristic Prediction

**Konstantin Schürholt**
konstantin.schuerholt@unisg.ch
AIML Lab, School of Computer Science
University of St.Gallen

**Dimche Kostadinov**
dimche.kostadinov@unisg.ch
AIML Lab, School of Computer Science
University of St.Gallen

**Damian Borth**
damian.borth@unisg.ch
AIML Lab, School of Computer Science
University of St.Gallen

## Abstract

Self-Supervised Learning (SSL) has been shown to learn useful and information-preserving representations. Neural Networks (NNs) are widely applied, yet their weight space is still not fully understood. Therefore, we propose to use SSL to learn *hyper-representations* of the weights of populations of NNs. To that end, we introduce domain specific data augmentations and an adapted attention architecture. Our empirical evaluation demonstrates that self-supervised representation learning in this domain is able to recover diverse NN model characteristics. Further, we show that the proposed learned representations outperform prior work for predicting hyper-parameters, test accuracy, and generalization gap as well as transfer to out-of-distribution settings. Code and datasets are publicly available[1].

## 1 Introduction

This work investigates populations of Neural Network (NN) models and aims to learn representations of them using Self-Supervised Learning. Within NN populations, not all model training is successful, i.e., some overfit and others generalize. This may be due to the non-convexity of the loss surface during optimization [Goodfellow et al., 2015], the high dimensionality of the optimization space, or the sensitivity to hyperparameters [Hanin and Rolnick, 2018], which causes models to converge to different regions in weight space. What is still not yet fully understood, is how different regions in weight space are related to model characteristics.

Previous work has made progress investigating characteristics of NN models, e.g by visualizing learned features [Zeiler and Fergus, 2014, Karpathy et al., 2015]. Another line of work compares the activations of pairs of NN models [Raghu et al., 2017, Morcos et al., 2018, Kornblith et al., 2019]. Both approaches rely on the expressiveness of the data, and are, in the latter case, limited to two models at a time. Other approaches predict model properties, such as accuracy, generalization gap, or hyperparameters from the margin distribution [Yak et al., 2019, Jiang et al., 2019], graph topology features [Corneanu et al., 2020] or eigenvalue decomposition of the weight matrices [Martin and Mahoney, 2019]. In a similar direction, other publications propose to investigate populations of models and to predict properties directly from their weights or weight statistics in a supervised way [Unterthiner et al., 2020, Eilertsen et al., 2020]. However, these manually designed features may not fully capture the latent model characteristics embedded in the weight space.

---

[1] https://github.com/HSG-AIML/NeurIPS_2021-Weight_Space_Learning

35th Conference on Neural Information Processing Systems (NeurIPS 2021).

| I. Model Zoo Generation | II. Representation Learning Approach | III. Down. Tasks |

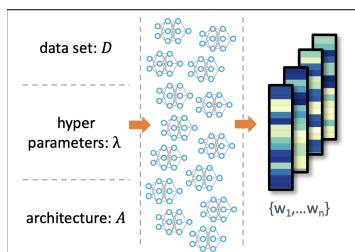 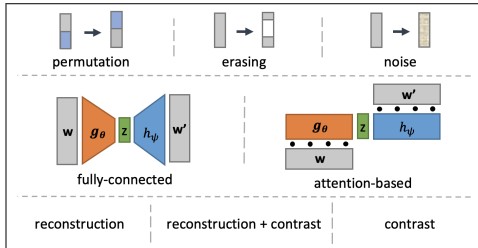 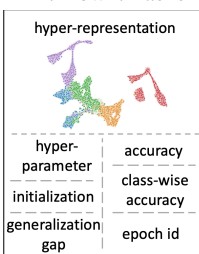

Figure 1: An overview of the proposed self-supervised representation learning approach. **I.** Populations of trained NNs form model zoos; each model is transformed in a vectorized form of its weights. **II.** Hyper-representations are learned from the model zoos using different augmentations, architectures, and Self-Supervised Learning tasks. **III.** Hyper-representations are evaluated on downstream tasks which predict model characteristics.

Therefore, our goal is to learn task-agnostic representations from populations of NN models able to reveal such characteristics. Self-Supervised Learning (SSL) is able to reveal latent structure in complex data without the need of labels, e.g., by compressing and reconstructing data [Goodfellow et al., 2016, Kingma and Welling, 2014]. Recently, a specific approach to SSL called contrastive learning has gained popularity [Misra and van der Maaten, 2019, Chen et al., 2020a,b, Grill et al., 2020]. Contrastive learning leverages inherent symmetries and equivariances in the data, allows to encode inductive biases and thus structure the learned representations.

In this paper, we propose a novel approach to apply SSL to learn representations of the weights of NN populations. We learn representations using reconstruction, contrast, and a combination of both. To that end, we adapt a transformer architecture to NN weights. Further, we propose three novel data augmentations for the domain of NN weights. We introduce structure preserving permutations of NN weights as augmentations, which make use of the structural symmetries within the NN weights that we find necessary for learning generalizing representations. We also adapt erasing [Zhong et al., 2020] and noise [Goodfellow et al., 2016] as augmentations for NN weights. We evaluate the learned representations by linear-probing for the generating factors and characteristics of the models. An overview for our learning approach is given in Figure 1.

To validate our approach, we perform extensive numerical experiments over different populations of trained NN models. We find that *hyper-representations*[2] can be learned and reveal the characteristics of the model zoo. We show that hyper-representations have high utility on tasks such as the prediction of hyper-parameters, test accuracy, and generalization gap. This empirically confirms our hypothesis on meaningful structures formed by NN populations. Furthermore, we demonstrate improved performance compared to the state-of-the-art for model characteristic prediction and outlay the advantages in out-of-distribution predictions. For our experiments, we use publicly available NN model zoos and introduce new model zoos. In contrast to [Unterthiner et al., 2020, Eilertsen et al., 2020], our zoos contain models with different initialization points and diverse configurations, and include densely sampled model versions during training. Our ablation study confirms that the various factors for generating a population of trained NNs play a vital role in how and which properties are recoverable for trained NNs. In addition, the relation between generating factors and model zoo diversity reveals that seed variation for the trained NNs in the zoos is beneficial and adds another perspective when recovering NNs' properties.

## 2 Model Zoos and Augmentations

**Model Zoo** We denote as $\mathcal{D}$ a data set that contains data samples with their corresponding labels. We denote as $\lambda$ the set of hyper-parameters used for training, (*e.g.*, loss function, optimizer, learning rate, weight initialization, batch-size, epochs). We define as $A$ the specific NN architecture. Training under different prescribed configurations $\{\mathcal{D}, \lambda, A\}$ results in a population of NNs which we refer to as *model zoo*. We convert the weights and biases of all NNs of each model into a vectorized form. In the resulting model zoo $\mathcal{W} = \{\mathbf{w}_1, ...., \mathbf{w}_M\}$, $\mathbf{w}_i$ denotes the flattened vector of dimension $N$, representing the weights and biases for one trained NN model.

---

[2]By *hyper-representation*, we refer to a learned representation from a population of NNs , i.e., a model zoo, in analogy to HyperNetworks [Ha et al., 2016], which are trained to generate weights for larger NN models.

**Augmentations**. Data augmentation generally helps to learn robust models, both by increasing the number of training samples and preventing overfitting of strong features [Shorten and Khoshgoftaar, 2019]. In contrastive learning, augmentations can be used to exploit domain-specific inductive biases, e.g., know symmetries or equivariances [Chen and Li, 2020]. To the best of our knowledge, augmentations for NN weights do not yet exist. In order to enable self-supervised representation learning, we propose three methods to augment individual instances of our model zoos.

Neurons in dense layers can change position without changing the overall mapping of the network, if in-going and out-going connections are changed accordingly [Bishop, 2006]. The relation between equivalent versions of the same network translates to permutations of incoming weights with matrix $\mathbf{P}$ and transposed permutation $\mathbf{P}^{\mathrm{T}}$ of the outgoing weights ($\mathbf{P}^{\mathrm{T}}\mathbf{P} = \mathbf{I}$). Considering the output at layer $l+1$, with weight matrices $\mathbf{W}$, biases $\mathbf{b}$, activations $\mathbf{a}$ and activation function $\sigma$, we have

$$\mathbf{z}^{l+1} = \mathbf{W}^{l+1}\sigma(\mathbf{W}^l\mathbf{a}^{l-1} + \mathbf{b}^l) + \mathbf{b}^{l+1} = \hat{\mathbf{W}}^{l+1}\sigma(\hat{\mathbf{W}}^l\mathbf{a}^{l-1} + \hat{\mathbf{b}}^l) + \mathbf{b}^{l+1}, \qquad (1)$$

where $\hat{\mathbf{W}}^{l+1} = \mathbf{W}^{l+1}(\mathbf{P}^l)^{\mathrm{T}}$, $\hat{\mathbf{W}}^l = \mathbf{P}^l\mathbf{W}^l$ and $\hat{\mathbf{b}}^l = \mathbf{P}^l\mathbf{b}^l$ are the permuted weight matrices and bias vector, respectively. The equivalences hold not only for the forward pass, but also for the backward pass and weight update.[3] The permutation can be extended to kernels of convolution layers. The *permutation augmentation* differs significantly from existing augmentation techniques. As an analogy, flips along the axis of images are similar, but specific instances from the set of possible permutations in the image domain. Each permutable layer with dimension $N_l$, has $N_l!$ different permutation matrices, and in total there are $\prod_l N_l!$ distinct, but equivalent versions of the same NN. While new data can be created by training new models, the generation is computationally expensive. The permutation augmentation, however, allows to compute valid NN samples at almost no computational cost. Empirically, we found the permutation augmentation crucial for our learning approach.

In computer vision and natural language processing, masking parts of the input has proven to be helpful for generalization [Devlin et al., 2019]. We adapt the approach of *random erasing* of sections in the vectorized forms of trained NN weights. As in [Zhong et al., 2020], we apply the erasing augmentation with a probability $p$ to an area that is randomly chosen with a lower and upper bounds $b_{low}$ and $b_{up}$. In our experiments, we set $p = 0.5$, $b_{low} = 0.03$, $b_{up} = 0.3$ and erase with zeros. Adding *noise augmentation* is another way of altering the exact values of NN weights without overly affecting their mapping, and has long been used in other domains [Goodfellow et al., 2016].

## 3 Hyper-Representation Learning

With this work, we propose to learn representations of structures in weight space formed by populations of NNs. We evaluate the representations by predicting model characteristics. A supervised learning approach has been demonstrated [Unterthiner et al., 2020, Eilertsen et al., 2020]. [Unterthiner et al., 2020] find that statistics of the weights (mean, var and quintiles) are superior to the weights to predict test accuracy, which we empirically confirm in our results. However, we intend to learn representations of the weights, that contain rich information beyond statistics. 'Labels' for NN models can be obtained relatively simply, yet they can only describe predefined characteristics of a model instance (e.g., accuracy) and so supervised learning may overfit few features, as [Unterthiner et al., 2020] show. Self-supervised approaches, on the other hand, are designed to learn task-agnostic representations, that contain rich and diverse information and are exploitable for multiple downstream tasks [LeCun and Misra, 2021]. Below, we present the used architectures and losses for the proposed self-supervised representation learning.

**Architectures and Self-Supervised Losses**. We apply variations of an encoder-decoder architecture. We denote the encoder as $g_\theta(\mathbf{w}_i)$, its parameters as $\theta$, and the hyper-representation with dimension $L$ as $\mathbf{z}_i = g_\theta(\mathbf{w}_i)$. We denote the decoder as $h_\psi(\mathbf{z}_i)$, its parameters as $\psi$, and the reconstructed NN weights as $\hat{\mathbf{w}}_i = h_\psi(\mathbf{z}_i) = h_\psi(g_\theta(\mathbf{w}_i))$. As is common in CL, we apply a projection head $p_\gamma(\mathbf{z}_i)$, with parameters $\gamma$, and denote the projected embeddings as $\bar{\mathbf{z}}_i = p_\gamma(\mathbf{z}_i) = p_\gamma(g_\theta(\mathbf{w}_i))$. In all of the architectures, we embed the hyper-representation $\mathbf{z}_i$ in a low dimensional space, $L < N$. We employ two common SSL strategies: reconstruction and contrastive learning. Autoencoders (AEs) with a reconstruction loss are commonly used to learn low-dimensional representations [Goodfellow

---

[3]Details, formal statements and proofs can be found in Appendix A.

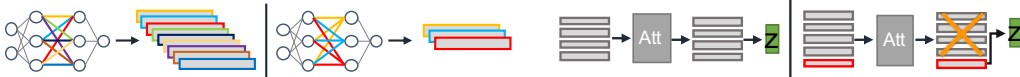

Figure 2: Trained NN weights are the input sequence to a transformer. **Left**. Each element in the sequence represents the weight that connects two neurons at two different layers. **Right**. Each element in the sequence represents the set of weights related to one neuron.

Figure 3: Multi-head attention-based encoder. **Left**. Regular sequence-to-sequence translation, each element of the output sequence is used. **Right**. An additional *compression token* is added to the sequence. From the output sequence, only the compression token is taken.

et al., 2016]. As AEs aim to minimize the reconstruction error, the representations attempt to fully encode samples. Further, contrastive learning is an elegant method to leverage inductive biases of symmetries and inductive biases [Bronstein et al., 2021].

**Reconstruction (ED):** For reconstruction, we minimize the MSE $\mathcal{L}_{MSE} = \frac{1}{M}\sum_{i=1}^{M}\|\mathbf{w}_i - h_\psi(g_\theta(\mathbf{w}_i))\|_2^2$. We denote the encoder-decoder with a reconstruction loss as ED.

**Contrast ($\mathbf{E}_c$):** For contrastive learning, we use the common NT_Xent loss [Chen et al., 2020a] as $\mathcal{L}_c$. For a batch of $M_B$ model weights, each sample is randomly augmented twice to form the two *views* $i$ and $j$. With the cosine similarity $\text{sim}(\bar{\mathbf{z}}_i, \bar{\mathbf{z}}_j) = \bar{\mathbf{z}}_i^T\bar{\mathbf{z}}_j/\|\bar{\mathbf{z}}_i\|\|\bar{\mathbf{z}}_j\|$, the loss is given as

$$\mathcal{L}_c = \sum_{(i,j)} -\log\frac{\exp(\text{sim}(\bar{\mathbf{z}}_i, \bar{\mathbf{z}}_j)/T}{\sum_{k=1}^{2M_B}\mathbb{I}_{k\neq i}\exp(\text{sim}(\bar{\mathbf{z}}_i, \bar{\mathbf{z}}_j)/T}, \tag{2}$$

where $\mathbb{I}_{k\neq i}$ is 1 if $k\neq i$ and 0 otherwise, and $T$ is the temperature parameter. We denote the encoder with a contrastive loss as $\mathrm{E}_c$.

**Reconstruction + Contrast ($\mathbf{E}_c\mathbf{D}$):** Further we combine reconstruction and contrast via $\mathcal{L} = \beta\mathcal{L}_{MSE} + (1-\beta)\mathcal{L}_c$ in order to achieve good quality compression via reconstruction and well-structured representations via the contrast. We denote this architecture with its loss as $\mathrm{E}_c\mathrm{D}$.

**Reconstruction + Positive Contrast ($\mathbf{E}_{c+}\mathbf{D}$):** In contrastive learning, many methods prevented mode collapse by using negative samples. The combined loss contains a reconstruction term $\mathcal{L}_{MSE}$, which can be seen as a regularizer that prevents mode collapse. Therefore, we also experiment with replacing $\mathcal{L}_c$ in our loss with a modified contrastive term without negative samples:

$$\mathcal{L}_{c+} = \sum_i -\log\left(\exp(\text{sim}(\bar{\mathbf{z}}_i^j, \bar{\mathbf{z}}_i^k))/T\right) = \sum_i -\text{sim}(\bar{\mathbf{z}}_i^j, \bar{\mathbf{z}}_i^k) + \log(T). \tag{3}$$

[Chen and He, 2021, Schwarzer et al., 2021] explore similar simplifications without reconstruction. We denote the encoder-decoder with the loss $\mathcal{L} = \beta\mathcal{L}_{MSE} + (1-\beta)\mathcal{L}_{c+}$ as $\mathrm{E}_{c+}\mathrm{D}$.

**Attention Module**. Our encoder and decoder pairs are symmetrical and of the same type. As there is no intuition on good inductive biases in the weight space, we apply fully connected feed-forward networks (FFN) as baselines. Further, wee use multi-head self-attention modules (Att) [Vaswani et al., 2017] as an architecture with very little inductive bias. In the multi-head self-attention module, we apply learned position encodings to preserve structural information [Dosovitskiy et al., 2020]. The explicit combination of value and position makes attention modules ideal candidates to resolve the permutation symmetries of NN weight spaces. We propose two methods to encode the weights into a sequence (Figure 2). In the first method, we encode each weight as a token in the input sequence. In the second method, we linearly transform the weights of one neuron or kernel and use it as a token. Further, we apply two variants to compress representations in the latent space (Figure 3). In the first variant, we aggregate the output sequence of the transformer and linearly compress it to a hyper-representation $\mathbf{z}_i$. In the second variant, similarly to [Devlin et al., 2019, Zhong et al., 2020], we add a learned token to the input sequence that we dub *compression token*. After passing the input sequence trough the transformer, only the compression token from the output sequence is linearly compressed to a hyper-representation $\mathbf{z}_i$. Without the compression token, the information is distributed across the output sequence. In contrast, the compression token is learned as an effective query to aggregate the relevant information from the other tokens, similar to [Jaegle et al., 2021]. The capacity of the compression token can be an information bottleneck. Its dimensionality is directly tied to the dimension of the value tokens and so its capacity affects the overall memory consumption.

| | | TETRIS-SEED | | | | | | | | | TETRIS-SEED | | | | |
|---|---|---|---|---|---|---|---|---|---|---|---|---|---|---|---|
| | Rec. | Eph | Acc | $F1_{C0}$ | $F1_{C1}$ | $F1_{C2}$ | $F1_{C3}$ | | Rec. | Eph | Acc | $F1_{C0}$ | $F1_{C1}$ | $F1_{C2}$ | $F1_{C3}$ |
| $E_c$ | - | 96.7 | **90.8** | 67.7 | 72.0 | **74.4** | 85.8 | FF | 0.0 | 80.0 | 85.3 | 57.3 | 53.9 | 64.5 | 80.1 |
| ED | **96.1** | 88.3 | 68.9 | 47.8 | 57.2 | 33.0 | 58.1 | $Att_W$ | 6.8 | 95.3 | 71.1 | 47.9 | 69.0 | 49.9 | 61.3 |
| $E_cD$ | 84.1 | **97.0** | 90.2 | **70.7** | **75.9** | 69.4 | **86.6** | $Att_{W+t}$ | 74.1 | 95.4 | 88.6 | 65.6 | 69.8 | **69.9** | **86.8** |
| $E_{c+}D$ | 95.9 | 94.0 | 69.9 | 48.9 | 58.1 | 32.5 | 58.8 | $Att_N$ | **89.4** | **97.1** | 88.4 | **71.4** | **80.8** | 69.1 | 82.3 |
| | | | | | | | | $Att_{N+t}$ | 84.1 | 97.0 | **90.2** | 70.7 | 75.9 | 69.4 | 86.6 |

Table 1: Ablation results over self-supervised learning losses. All models implemented with attention-based reference architecture. All values are given in %.

Table 2: Ablation results in % under $E_cD$ setup. We use feed-forward FF and attention-based variants with weight and neuron encoding $Att_W$ and $Att_N$ each with $+t$ and without compress. token.

**Downstream Tasks**. We use linear probing [Grill et al., 2020] as a proxy to evaluate the utility of the learned hyper-representations. As downstream tasks we use accuracy prediction (Acc), generalization gap (GGap), epoch prediction (Eph) as proxy to model versioning, F1-Score prediction ($F1_C$), learning rate (LR), $\ell_2$-regularization ($\ell_2$-reg), dropout (Drop) and training data fraction (TF). Using such targets, we solve a regression problem and measure the $R^2$ score [Wright, 1921]. We also evaluate for hyper-parameters prediction tasks, like the activation function (Act), optimizer (Opt), initialization method (Init). Here, we train a linear perceptron by minimizing a cross entropy loss [Goodfellow et al., 2016] and measure the prediction accuracy.

# 4 Empirical Evaluation

## 4.1 Model Zoos

**Publicly Available Model Zoos**. [Unterthiner et al., 2020] introduced model zoos of CNNs with 4970 parameters trained on MNIST [LeCun and Cortes], Fashion-MNIST [Xiao et al., 2017], CIFAR10 [Krizhevsky et al.] and SVHN [Netzer et al., 2011], and made them available under CC BY 4.0. We refer to these as `MNIST-HYP`, `FASHION-HYP`, `CIFAR10-HYP` and `SVHN-HYP`. We categorize these zoos as *large* due to their number of parameters. In their model zoo creation, the CNN architecture and seed were fixed, while the activation function, initialization method, optimizer, learning rate, $\ell_2$ regularization, dropout and the train data fraction were varied between the models.

| | | TETRIS-SEED | | | | | | |
|---|---|---|---|---|---|---|---|---|
| | - | P | E | N | P,E | P,N | E,N | P,E,N |
| ED | 88.1 | **95.5** | 89.8 | 88.8 | **96.1** | 95.6 | 89.7 | **96.1** |
| $E_cD$ | 49.2 | **72.9** | 67.3 | 59.4 | **84.1** | 81.9 | 65.4 | **84.1** |
| $E_{c+}D$ | 86.3 | **95.6** | 88.5 | 87.6 | **95.9** | 95.8 | 89.0 | **96.0** |

Table 3: Ablation results for different augmentations and representation learning tasks. We use: permutation (P), erasing (E) and noise (N) augmentation and report test-split reconstruction $R^2$ scores in %.

**Our Model Zoos**. We hypothesize that using only one fixed seed may limit the variation in characteristics of a zoo. To address that, we train zoos where we also vary the seed and perform an ablation study below. As a toy example to test architectures, SSL tasks and augmentations, we first create a 4x4 grey-scaled image data set that we call *tetris* by using four tetris shapes.

We introduce two zoos, which we call `TETRIS-SEED` and `TETRIS-HYP`, which we group under *small*. Both zoos contain FFN with two layers and have a total number of 100 learnable parameters. In the `TETRIS-SEED` zoo, we fix all hyper-parameters and vary only the seed to cover a broad range of the weight space. The `TETRIS-SEED` zoo contains 1000 models that are trained for 75 epochs. To enrich the diversity of the models, the `TETRIS-HYP` zoo contains FFNs, which vary in activation function [`tanh`, `relu`], the initialization method [`uniform`, `normal`, `kaiming normal`, `kaiming uniform`, `xavier normal`, `xavier uniform`] and the learning rate [$1e$-3, $1e$-4, $1e$-5]. In addition, each combination is trained with seeds 1-100. Out of the 3600 models in total, we have successfully trained 2900 for 75 epochs - the remainders crashed and are disregarded. Similarly to `TETRIS-SEED`, we further create zoos of CNN models with 2464 parameters, each using the MNIST and Fashion-MNIST data sets, called `MNIST-SEED` and `FASHION-SEED` and grouped them as *medium*. To maximize the coverage of the weight space, we again initialize models with seeds 1-1000. [4]

---

[4]Full details on the generation of the zoos can be found in the Appendix Section C

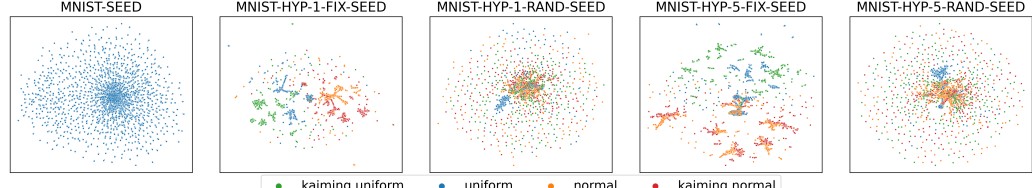

Figure 4: UMAP dimensionality reduction for NN model zoos created with different generating factors. Colors represent initialization methods. Compare numerical results in Table 4.

| | MNIST-SEED | | | MNIST-HYP-1-FIX-SEED | | | MNIST-HYP-1-RAND-SEED | | | MNIST-HYP-5-FIX-SEED | | | MNIST-HYP-5-RAND-SEED | | |
|---|---|---|---|---|---|---|---|---|---|---|---|---|---|---|---|
| VAR | .234 | | | .155 | | | .152 | | | .091 | | | .092 | | |
| $VAR_c$ | .234 | | | .101 | | | .164 | | | .094 | | | .100 | | |
| $R_{tr}$ | 73.4 | | | 91.3 | | | 80.0 | | | 81.3 | | | 65.6 | | |
| $R_{tes}$ | 59.0 | | | 72.9 | | | 37.2 | | | 70.0 | | | 38.6 | | |
| | W | s(W) | $E_{c+}$D | W | s(W) | $E_c$D | W | s(W) | $E_{c+}$D | W | s(W) | $E_c$D | W | s(W) | $E_c$D |
| EPH | 84.5 | **97.7** | 97.3 | 07.2 | 06.2 | **11.6** | -34 | 00.5 | **04.1** | -2.7 | 7.4 | **10.2** | -14 | **09.6** | 07.2 |
| ACC | 91.3 | 98.7 | **98.9** | 73.6 | 72.9 | **89.5** | -06 | 60.1 | **85.7** | 57.7 | 75.1 | **79.4** | -13 | 74.5 | **82.5** |
| GGAP | 56.9 | 66.2 | **66.7** | 55.3 | 42.1 | **61.9** | -36 | 31.1 | **54.9** | 37.8 | 36.7 | **57.7** | -3.2 | 46.1 | **60.5** |
| INIT | – | – | – | **87.7** | 63.3 | 75.4 | 38.3 | **56.1** | 48.2 | **80.6** | 54.5 | 51.4 | 35.5 | **47.2** | 40.4 |

Table 4: $R^2$ score in %. Results on the impact of the generating factors for the model zoos. **Top:** VAR is the variance of the weights. $VAR_c$ denotes the mean of the variances for groups of samples with shared initialization method and activation function. $R_{tr}$ and $R_{tes}$ are the reconstruction $R^2$ of train and test split on a reference $E_{c+}$D architecture trained for 500 epochs. **Bottom:** Downstream task performance compared to baselines predicting epochs, accuracy, generalization gap and initialization.

**Model Zoo Generating Factors**. Prior work discusses the impact of random seeds on properties of model zoos. While [Yak et al., 2019] use multiple random seeds for the same hyper-parameter configuration, [Unterthiner et al., 2020] explicitly argue against that to prevent information leakage between samples. To disentangle the generating factors (seeds and hyper-parameters) and model properties, we have created five zoos with approximately the same number of CNN models trained on MNIST [4]. MNIST-SEED varies only the random seed (1-1000), MNIST-HYP-1-FIX-SEED varies the hyper-parameters with one fixed seed per configuration (similarly to [Unterthiner et al., 2020]). To decouple the hyper-parameter configuration from one specific seed, MNIST-HYP-1-RAND-SEED draws 1 random seeds for each hyper-parameter configuration. To investigate the influence of repeated configurations, in MNIST-HYP-5-FIX-SEED and MNIST-HYP-5-RAND-SEED for each hyper-parameter configurations we add 5 models with different seeds, either 5 fixed seeds or randomly drawn seeds.

## 4.2 Training and Testing Setup

**Architectures**. We evaluate our approach with different types of architectures, including $E_c$, ED, $E_c$D and $E_{c+}$D as detailed in Section 3. The encoder E and decoders D in the FFN baseline are symmetrical 10 [FC-ReLU]-layers each and linearly reduce dimensionality for the input to the latent space **z**. Considering the attention-based encoder and decoder, on the TETRIS-SEED and TETRIS-HYP zoos, we used 2 attention blocks with 1 attention head each, token dimensions of 128 and FC layers in the attention module of dimension 512. On the larger zoos, we use up to 4 attention heads in 4 attention blocks, token dimensions of up to size 800 and FC layers in the attention module of dimension 1000. For the combined losses, we evaluate different $\beta \in [0.05, 0.85]$.

**Hyper-representation Learning and Downstream Tasks**. We apply the proposed data augmentation methods for representation learning (see Section 2). We run our representation learning algorithms for up to 2500 epochs, using the adam optimizer [Kingma and Ba, 2014], a learning rate of 1e-4, weight decay of 1e-9, dropout of 0.1 percent and batch-sizes of 500. In all of our experiments, we use 70% of the model zoos for training, 15% for validation and 15% for testing. We use checkpoints of all epochs, but ensure that samples from the same models are either in the

| | TETRIS-SEED | | | | | | | | TETRIS-HYP | | | | | | |
|---|---|---|---|---|---|---|---|---|---|---|---|---|---|---|---|
| | W | $PCA_l$ | $PCA_c$ | $PCA_r$ | $U_m$ | s(W) | $E_cD$ | | W | $PCA_l$ | $PCA_c$ | $PCA_r$ | $U_m$ | s(W) | $E_cD$ |
| Eph | 55.4 | 46.9 | 77.8 | 93.5 | 0.00 | 96.4 | **97.0** | Eph | 02.8 | 02.7 | 0.04 | 13.1 | 0.03 | 16.3 | **16.7** |
| Acc | 49.1 | 23.9 | 74.7 | 74.9 | 0.01 | 86.9 | **90.2** | Acc | 11.0 | 14.0 | 0.81 | 73.8 | 0.7 | 80.2 | **83.7** |
| Ggap | 43.9 | 28.3 | 72.9 | 75.7 | 0.01 | 81.0 | **81.9** | GGap | 12.9 | 12.9 | 0.31 | 76.0 | 0.3 | 79.2 | **81.6** |
| $F_{C0}$ | 26.6 | 0.07 | 52.8 | 49.1 | 0.02 | 62.5 | **70.7** | LR | -4.7 | -3.2 | -1.3 | -0.4 | -1.4 | **53.4** | 51.1 |
| $F_{C1}$ | 41.4 | 30.2 | 48.6 | 58.6 | 0.01 | 63.1 | **75.9** | Act | 74.7 | 72.7 | 45.1 | 74.2 | 45.9 | 73.6 | **86.9** |
| $F_{C2}$ | 41.5 | 0.06 | 38.3 | 38.5 | 0.01 | 60.9 | **69.4** | Init | 38.5 | 36.4 | 32.7 | 35.4 | 33.6 | 46.6 | **47.1** |
| $F_{C3}$ | 53.9 | 44.6 | 73.8 | 72.2 | 0.00 | 75.2 | **86.6** | | | | | | | | |

Table 5: $R^2$ given in %. **Left**. Reconstruction, epoch, accuracy, generalization gap and class-wise F-scores prediction. **Right**. Epoch, accuracy, generalization gap, learning rate, seed, activation function and initialization prediction.

train, validation or the test split of the zoo. As quality metric for the hyper-representation learning, we track the reconstruction $R^2$ on the validation split of the zoo and report the reconstruction $R^2$ on the test split. As a proxy for how much useful information is contained in the hyper-representation, we evaluate on downstream tasks as described in Section 3. To ensure numerical stability of the solution to the linear probing, we apply Tikhonov regularization [Tikhonov and Arsenin, 1977] with regularization parameter $\alpha$ in the range [$1e$-5, $1e3$] (we choose $\alpha$ by cross-validating over the $R^2$ score of the validation split of the zoo) and report the $R^2$ score of the test split of the zoo. To minimize the cross entropy loss for the categorical hyper-parameter prediction, we use the adam optimizer with learning rate of $1e$-4 and weight-decay of $1e$-6. The linear probing is applied to the same train-validation-test splits as it is in our representation learning setup.

**Out-of-Distribution Experiments**. We follow a setup for out-of-distribution experiments similar to [Unterthiner et al., 2020]. We investigate how well the linear probing estimator computed over hyper-representations generalize to yet unseen data. Therefore, we use the zoos `MNIST-HYP`, `FASHION-HYP`, `CIFAR10-HYP` and `SVHN-HYP`. On each zoo, we apply our self-supervised approach to learn their corresponding hyper-representations and fit a linear probing estimator to each of them (in-distribution). We then apply both the hyper-representation mapper and the linear probing estimator of one zoo on the weights of the other zoos (out-of-distribution). The target ranges and distributions vary between the zoos. Linear probe prediction may preserve the relation between predictions, but include a bias. Therefore, we use Kendall's $\tau$ coefficient as performance metric, which is a measure of rank correlation. It measures the ordinal association between two measured quantities [Kendall, 1938].

**Baselines, Computing Infrastructure and Run Time**. As baseline, we use the model weights (W). In addition, we also consider PCA with linear ($PCA_l$), cosine ($PCA_c$), and radial basis kernel ($PCA_r$), as well as UMAP ($U_m$) [McInnes et al., 2018]. We further compare to layer-wise weight-statistics (mean, var, quintiles) s(W) as in [Unterthiner et al., 2020, Eilertsen et al., 2020]. As computing hardware, we use half of the available resources from NVIDIA DGX2 station with 3.3GHz CPU and 1.5TB RAM memory, that has a total of 16 1.75GHz GPUs, each with 32GB memory. To create one small and medium zoo, it takes 1 to 2 days and 10 to 12 days, respectively. For one experiment over the small zoo it takes around 3 hours to learn the hyper-representation on a single GPU and evaluate on the downstream tasks. It takes approximately 1 day for the medium zoos and 2 to 3 days for the large scale zoos for the same experiment. The representation learning model size ranges from 225k on the small zoos to 65M parameters on the largest zoos. We use ray.tune [Liaw et al., 2018] for hyperparameter optimization and track experiments with W&B [Biewald, 2020].

### 4.3 Results

**Augmentation Ablation**. To evaluate the impact of the proposed augmentations (Section 2) for our representation learning method, we present an ablation analysis on the `TETRIS-SEED` zoo, in which we measure $R^2$ for ED, $E_cD$ and $E_{c+}D$[5]. We use 120 permutations, a probability of 0.5 for erasing the weights, and zero-mean noise with standard deviation 0.05 (see Table 3). We find the permutation augmentation to be necessary for generalization - particularly under higher compression ratios. The additional samples generated with the permutation appear to effectively prevent overfitting

---

[5]We leave out $E_c$ as it does not use a reconstruction loss

| | MNIST-HYP | | | FASHION-HYP | | | CIFAR10-HYP | | | SVHN-HYP | | |
|---|---|---|---|---|---|---|---|---|---|---|---|---|
| | W | s(W) | $E_cD$ | W | s(W) | $E_cD$ | W | s(W) | $E_cD$ | W | s(W) | $E_cD$ |
| EPH | 25.8 | 33.2 | **50.0** | 26.6 | 34.6 | **51.3** | 25.7 | 30.3 | **53.3** | 22.8 | 37.8 | **52.6** |
| ACC | 74.7 | 81.5 | **94.9** | 70.9 | 78.5 | **96.2** | 76.4 | 82.9 | **92.7** | 80.5 | 82.1 | **91.1** |
| GGAP | 23.4 | 24.4 | **27.4** | 48.1 | 41.1 | **49.0** | 37.7 | 37.4 | **40.4** | 38.7 | 42.2 | **44.2** |
| LR | 29.3 | 34.3 | **37.1** | 33.5 | 35.6 | **42.4** | 27.4 | 32.3 | **44.7** | 24.5 | 33.4 | **49.1** |
| $\ell_2$-REG | 12.5 | 16.5 | **20.1** | 11.9 | 16.3 | **25.0** | 08.7 | 13.8 | **28.0** | 09.0 | 13.6 | **28.0** |
| DROP | 28.5 | 19.2 | **35.8** | 26.7 | 21.3 | **38.3** | 16.7 | 16.5 | **33.8** | 09.0 | 14.6 | **23.3** |
| TF | 03.8 | 07.8 | **15.9** | 08.1 | 08.2 | **22.1** | 08.4 | 06.9 | **35.4** | 03.2 | 08.8 | **21.4** |
| ACT | 88.6 | 81.1 | **88.7** | 89.8 | 82.4 | **90.1** | 88.3 | 80.3 | **90.0** | 86.9 | 78.8 | **87.2** |
| INIT | **94.6** | 72.0 | 80.6 | **95.7** | 76.5 | 86.7 | **93.5** | 73.3 | 82.6 | **91.0** | 73.0 | 82.8 |
| OPT | **76.7** | 65.4 | 66.4 | **79.9** | 67.4 | 73.0 | **74.0** | 65.5 | 71.0 | **72.5** | 68.2 | 72.3 |

Table 6: **Top 7 Rows**. $R^2$ score in % for Eph, Acc, GGap, LR $\ell_2$-reg, Drop and TF prediction. **Bottom 3 Rows**. Accuracy score for Act, Init and Opt prediction.

of the training set. Without the permutation augmentation, the test performance diverges after few training epochs. Erasing further improves test performance and allows for extended training without overfitting. The addition of noise yields inconsistent results and is difficult to tune, so, we omit it in our further experiments.

**Architecture Ablation**. The different architectures are compared in Table 1. The results show that within the set of used NN architectures for hyper-representation learning, the attention-based architectures learn considerably faster, yield lower reconstruction error and have the highest performance on the downstream tasks compared to the FFN-based architectures. We attribute this to the attention modules, which are able to reliably capture long-range relations on complex data due the global field of visibility in each layer. While tokenizing each weight individually ($Att_W$) is able to learn, the computational load is significant, even for a small zoo, due to the large number of tokens in the sequences. The memory load prevents the application of that encoding on larger zoos. We find $Att_{N+t}$ embedding all weights of one neuron (or convolutional kernel) to one token in combination with compression tokens shows the overall best performance and scales to larger architectures. Compression tokens achieve higher performance, which we confirm on larger and more complex datasets. The dedicated token gathers information from all other tokens of the sequence in several attention layers. This appears to enable the hyper-representation to grasp more relevant information than linearly compressing the entire sequence. On the other hand, compression tokens are only an advantage, if their capacity is high enough, in particular higher than the bottleneck.

**Self-Supervised Learning Ablation**. In Table 2, we evaluate the usefulness of the self-supervised learning tasks (Section 3). The application of purely contrastive loss in $E_c$, learns very useful representations for the downstream tasks. However, $E_c$ heavily depends on expressive projection heads and by design cannot reconstruct samples to further investigate the representation. Pure reconstruction ED results in embeddings with a low reconstruction loss, but comparably low performance on downstream tasks. Among our losses, the combination of reconstruction with contrastive loss as in $E_cD$, provides hyper-representations $\mathbf{z}$ that have the best overall performance. The variation $E_{c+}D$, is closer to ED in general performance, but still outperforms it on the downstream tasks. The addition of a contrastive loss with projection head helps to pronounce distinctiveness, so that the hyper-representations are good at reconstructing the NN weights, and in revealing properties of the NNs through the encoder. Empirically, we found that on some of the larger zoos, $E_{c+}D$ performed better than $E_cD$. On these zoos, it appears that $E_{c+}D$ is most suitable for homogeneous zoos without distinct clusters, while $E_cD$ is suitable for zoos with more sub-structures, compare Figure 4 and Table 4. We therefore applied both learning strategies on all zoos and report the more performant one.

**Zoo Generating Factors Ablation**. Figure 4 visualizes the weights of the zoos, which contain models of all epochs[6]. In Table 4, we report numerical properties. Only changing the seeds appears to result in homogeneous development with very high correlation between $s(W)$ and the properties of the samples in the zoo, as previous work already indicated [Schürholt and Borth, 2021]. Varying the hyper-parameters reduces the correlation. With fixed seeds, we observe clusters of models with shared initialization method and activation function. Quantitatively we obtain lower $VAR_c$ and high

---

[6]Further visualizations can be found in the Appendix Section C

| | MNIST-HYP | | | FASHION-HYP | | | SVHN-HYP | | | CIFAR10-HYP | | |
|---|---|---|---|---|---|---|---|---|---|---|---|---|
| | W | s(W) | $E_{c+}D$ | W | s(W) | $E_{c+}D$ | W | s(W) | $E_{c+}D$ | W | s(W) | $E_{c+}D$ |
| MNIST-HYP | **.36** | .29 | **.36** | .21 | .14 | **.27** | **.26** | .12 | .23 | -.01 | -.04 | **.02** |
| FASHION-HYP | -.02 | **.08** | .02 | .54 | .48 | **.56** | .06 | **.14** | .01 | .07 | .10 | **.27** |
| SVHN-HYP | .05 | **.15** | -.04 | -.02 | **.27** | .10 | .44 | .34 | **.45** | -.02 | .08 | **.10** |
| CIFAR10-HYP | **.11** | .09 | .06 | .38 | .36 | **.39** | .14 | .14 | **.15** | **.41** | .28 | .35 |

Table 7: Kendall's $\tau$ score for the generalization gap (GGap) prediction. We train estimators for each zoo (rows) and evaluate on all zoos (columns). The block diagonal elements contain the in-distribution prediction values. The remaining values are for out-of-distribution prediction.

predictive value of $W$ for the initialization method. That seems a plausible outcome, given that the architecture and activation function determines the shape of the loss surface, while the seed and initialization method decide the starting point. Such clustering appear to facilitate the prediction of categorical characteristics from the weights. We observe similar properties in the zoos of [Unterthiner et al., 2020], see Appendix Section C. Initializing models with random seeds disperses the clusters, compare Figure 4 and Table 4. While VAR between 1 fixed and 1 random seed is comparable, $VAR_c$ is considerably smaller with fixed seed, the predictive value of $W$ for the initialization methods drops significantly. Random seeds also appear to make both the reconstruction as well as NN property prediction more difficult. The repetition of configurations with five seeds hinders shortcuts in the weight space, make reconstruction and the prediction of characteristics harder. Thus, we conclude that changing only the seeds results in models with very similar evolution during learning. In these zoos, statistics of the weights perform best. In contrast, using one seed shared across models might create shortcuts in the weight space. Zoos that vary both appear to be most diverse and hardest for learning and NN property prediction. Across all zoos with hyperparameter changes, learned hyper-representations significantly outperform the baselines.

**Downstream Tasks**. We learn and evaluate our hyper-representations on 11 different zoos: TETRIS-SEED, TETRIS-HYP, 5 variants of MNIST, MNIST-HYP, FASHION-HYP, CIFAR10-HYP and SVHN-HYP. We compare to multiple baselines and s(W). The results are shown in Tables 4, 5 and 6. On all model zoos, hyper-representations learn useful features for the downstream tasks, which outperform the actual NN weights and biases, all of the baseline dimensionality reduction methods as well as s(W) [Unterthiner et al., 2020]. On the TETRIS-SEED, and MNIST-SEED model zoos, s(W) achieves high $R^2$ scores on all downstream tasks (see Table 5 left). As discussed above, these zoos contain a strong correlation between $s(W)$ and sample properties. Nonetheless, learned hyper-representations achieve higher $R^2$ scores on all characteristics, with the exception of MNIST-SEED Eph, where $E_{c+}D$ is competitive to s(W). On TETRIS-HYP, the overall performance of all methods is lower compared to TETRIS-SEED (see Table 5 right), making it the more challenging zoo. Here, too, hyper-representations have the highest $R^2$ score on all downstream tasks, except for the LR prediction. On MNIST-HYP, FASHION-HYP, CIFAR10-HYP and SVHN-HYP, hyper-representations outperform s(W) on all downstream tasks and achieve higher $R^2$ score compared to W on the prediction of continuous hyper-parameters and activation prediction. On the remaining categorical hyper-parameters initialization method and optimizer, the weight space achieves the highest $R^2$ scores (see Table 6). We explain this with the small amount of variation in the model zoo (see Section 4.1), which allows to separate these properties in weight space.

**Out-of-Distribution Prediction**. Table 7 shows the out-of-distribution results for generalization gap prediction[7], which is a very challenging task. The task is rendered more difficult by the fact that many of the samples which have to be ordered by their performance are very close in performance. As the results show, hyper-representations are able to recover the order of samples by performance to a significant degree. Further, hyper-representations outperform the baselines in Kendall's $\tau$ measure in the majority of the results. This verifies that our approach indeed preserves the distinctive information about trained NNs while compactly relating to their common properties, including the characteristics of the training data. Presumably, learning representations on zoos with multiple different datasets would improve the out-of-distribution capabilities even further.

---

[7]In Appendix Section D, we also give results for other tasks, including epoch id and test accuracy prediction

## 5   Related Work

There is ample research evaluating the structures of NNs by visualizing activations, *e.g.*, [Zeiler and Fergus, 2014, Karpathy et al., 2015, Yosinski et al., 2015], which allow some insights in the patterns of, *e.g.*, the kernels of CNNs. Other research evaluated networks by computing a degree of similarity between networks. [Laakso and Cottrell, 2000] compared the activations of NNs by a measure of "sameness". [Li et al., 2015] computed correlations between the activations of different nets. [Wang et al., 2018] tried to match the subspaces of the activation spaces in different networks [Johnson, 2019], which showed to be unreliable. [Raghu et al., 2017, Morcos et al., 2018, Kornblith et al., 2019] applied correlation metrics to NN activations in order to study the learning dynamics and compare NNs. [Dinh et al., 2017] link model properties to characteristics of the loss surface around the minimum solution. In contrast comparing models with similarity metrics, other efforts map models to an absolute representation. [Jia et al., 2019] approximated the space of DNN activations with a convex hull. [Jiang et al., 2019] also used activations to approximate the margin distribution and predict the generalization gap. [Corneanu et al., 2020] proposed persistent homology by using a connectivity patterns in the NN activation, and compute topological summaries.

While previously mentioned related work studied measures defined on the activations for insights about the NN characteristics, the methods applied on populations of NN weights have not received much attention for the same purpose. [Martin and Mahoney, 2019] relate the empirical spectral density of the weight matrices to accuracy, indicating that the weight space alone contains relevant information about the model. [Eilertsen et al., 2020] evalutaed a classifier for hyper-parameter prediction directly from the weights. In contrast, we learn a general purpose hyper-representations in self-supervised fashion. [Unterthiner et al., 2020] proposed layer-wise statistics derived from the NN models to predict the test accuracy of NN model. In [Schürholt and Borth, 2021], subspaces of the weights space of populations of NNs are investigated for model uniqueness and epoch order.

In the proposed work, we model associations between the different weights in NN using an attention-based module. This helps us to learn representations that compactly extract the relevant and meaningful information while considering the correlations between the weights in an NN.

## 6   Conclusions

In this work, we present a novel approach to learn hyper-representations from the weights of neural networks. To that end, we proposed novel augmentations, self-supervised learning losses and adapted multi-head attention-based architectures with suitable weight encoding for this domain. Further, we introduced several new model zoos and investigated their properties. We showed that not only learned neural networks but also their hyper-representations contain the latent footprint of their training data. We demonstrated high performance on downstream tasks, exceeding existing methods in hyper-parameters, test accuracy, and generalization gap prediction and showing the potential in an out-of-distribution setting.

**Acknowledgements**   Leading up to this paper, there were many discussions with colleagues, which we would like to acknowledge. We are particularly grateful to Marco Schreyer, Xavier Giró-i-Nieto, Pol Caselles Rico and Diyar Taskiran.

**Funding Disclosure**   This work is supported by the University of St.Gallen Basic Research Fund.

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
