# Self-Supervised Representation Learning on Neural Network Weights for Model Characteristic Prediction −Appendix

In the following, we provide the Appendix as part of the supplementary material to the main paper. In Appendix Section A, we prove the permutation equivalence in the forward and backward pass for (i) fully connected and (ii) convolutional layers. We also include an empirical evaluation of the equivalence. In Section B, we provide further information about the downstream tasks and their targets. Section C contains additional content about the model zoos. It includes high-level information about the zoo's properties, the modes of variation between the samples in the zoos, and the exact architectures used when generating a zoo of trained neural networks. We also provide visualizations of some of the properties of our model zoo for better intuition. Lastly, Section D includes additional ablations and results, both in an in-distribution and out-of-distribution settings.

# A. Permutation Augmentation

In this appendix section, we give the full derivation about the permutation equivalence in the proposed permutation augmentation (Section 2 in the paper). In the following appendix subsections, we show the equivalence in the *forward* and *backward* pass through the neural network with original learnable parameters and the permutated neural network.

## A.1 Neural Networks and Back-propagation

Consider a common, fully-connected feed-forward neural network (FFN). It maps inputs $\mathbf{x} \in \mathbb{R}^{N_0}$ to outputs $\mathbf{y} \in \mathbb{R}^{N_L}$. For a FFN with $L$ layers, the forward pass reads as

$$
\begin{aligned}
\mathbf{a}^0 &= \mathbf{x}, \\
\mathbf{n}^l &= \mathbf{W}^l \mathbf{a}^{l-1} + \mathbf{b}^l, \qquad l \in \{1, \cdots, L\}, \\
\mathbf{a}^l &= \sigma(\mathbf{n}^l), \qquad l \in \{1, \cdots, L\}.
\end{aligned}
\tag{1}
$$

Here, $\mathbf{W}^l \in \mathbb{R}^{N_l \times N_{l-1}}$ is the weight matrix of layer $l$, $\mathbf{b}^l$ the corresponding bias vector. Where $N_l$ denotes the dimension of the layer $l$. The activation function is denoted by $\sigma$, it processes the layer's weighted sum $\mathbf{n}^l$ to the layer's output $\mathbf{a}^l$.

Training of neural networks is defined as an optimization against a objective function on a given dataset, *i.e.* their weights and biases are chosen to minimize a cost function, usually called *loss*, denoted by $\mathcal{L}$. The training is commonly done using a gradient based rule. Therefore, the update relies on the gradient of $\mathcal{L}$ with respect to weight $\mathbf{W}^l$ and the bias $\mathbf{b}^l$, that is it relies on $\nabla_{\mathbf{W}} \mathcal{L}$ and $\nabla_{\mathbf{b}} \mathcal{L}$, respectively. Back-propagation facilitates the computation of these gradients, and makes use of the chain rule to back-propagate the prediction error through the network [Rumelhart et al., 1986]. We express the error vector at layer $l$ as

$$
\delta^l = \nabla_{\mathbf{n}^l} \mathcal{L},
\tag{2}
$$

and further use it to express the gradients as

$$
\begin{aligned}
\nabla_{\mathbf{W}^l} \mathcal{L} &= \delta^l (\mathbf{a}^{l-1})^{\mathrm{T}}, \\
\nabla_{\mathbf{b}^l} \mathcal{L} &= \delta^l.
\end{aligned}
\tag{3}
$$

The output layer's error is simply given by

$$
\delta^L = \nabla_{\mathbf{a}^L} \mathcal{L} \odot \sigma'(\mathbf{n}^L),
\tag{4}
$$

where $\odot$ denotes the Hadamard element-wise product and $\sigma'$ is the activation's derivative with respect to its argument. Subsequent earlier layer's error are computed with

$$
\delta^l = (\mathbf{W}^{l+1})^{\mathrm{T}} \delta^{l+1} \odot \sigma'(\mathbf{n}^l), \qquad l \in \{1, \cdots, L-1\}.
\tag{5}
$$

A usual parameter update takes on the form

$$
(\mathbf{W}^l)_{\text{new}} = \mathbf{W}^l - \beta \nabla_{\mathbf{W}^l} \mathcal{L},
\tag{6}
$$

where $\beta$ is a positive learning rate.

## A.2 Proof: Permutation Equivalence

In the following appendix subsection, we show the permutation equivalence for feed-forward and convolutional layers.

**Permutation Equivalence for Feed-forward Layers** Consider the permutation matrix $\mathbf{P}^l \in \mathbb{N}^{N_i \times N_l}$, such that $(\mathbf{P}^l)^{\mathrm{T}} \mathbf{P}^l = \mathbf{I}$, where $\mathbf{I}$ is the identity matrix. We can write the weighted sum for layer $l$ as

$$
\begin{aligned}
\mathbf{n}^{l+1} &= \mathbf{W}^{l+1} \, \mathbf{a}^l + \mathbf{b}^{l+1} \\
&= \mathbf{W}^{l+1} \, \sigma(\mathbf{n}^l) + \mathbf{b}^{l+1} \\
&= \mathbf{W}^{l+1} \, \sigma(\mathbf{W}^l \mathbf{a}^{l-1} + \mathbf{b}^l) + \mathbf{b}^{l+1}.
\end{aligned}
\tag{7}
$$

As $\mathbf{P}^l$ is a permutation matrix and since we use the element-wise nonlinearity $\sigma(.)$, it holds that

$$\mathbf{P}^l \sigma(\mathbf{n}^l) = \sigma(\mathbf{P}^l \mathbf{n}^l), \tag{8}$$

which implies that we can write

$$
\begin{aligned}
\mathbf{n}^{l+1} &= \mathbf{W}^{l+1}\, \mathbf{I}\, \sigma(\mathbf{W}^l\, \mathbf{a}^{l-1} + \mathbf{b}^l) + \mathbf{b}^{l+1} \\
&= \mathbf{W}^{l+1}\, (\mathbf{P}^l)^{\mathrm{T}}\, \mathbf{P}^l\, \sigma(\mathbf{W}^l\, \mathbf{a}^{l-1} + \mathbf{b}^l) + \mathbf{b}^{l+1} \\
&= \mathbf{W}^{l+1}\, (\mathbf{P}^l)^{\mathrm{T}}\, \sigma(\mathbf{P}^l\, \mathbf{W}^l\, \mathbf{a}^{l-1} + \mathbf{P}^l\, \mathbf{b}^l) + \mathbf{b}^{l+1} \\
&= \hat{\mathbf{W}}^{l+1}\, \sigma(\hat{\mathbf{W}}^l\, \mathbf{a}^{l-1} + \hat{\mathbf{b}}^l) + \mathbf{b}^{l+1},
\end{aligned}
\tag{9}
$$

where $\hat{\mathbf{W}}^{l+1} = \mathbf{W}^{l+1}(\mathbf{P}^l)^{\mathrm{T}}$, $\hat{\mathbf{W}}^l = \mathbf{P}^l\mathbf{W}^l$ and $\hat{\mathbf{b}}^l = \mathbf{P}^l\mathbf{b}^l$ are the permuted weight matrices and bias vector.

Note that rows of weight matrix and bias vector of layer $l$ are exchanged together with columns of the weight matrix of layer $l + 1$. In turn, $\forall l \in \{1, \cdots, L - 1\}$, (9) holds true. At any layer $l$, there exist $N_l$ different permutation matrices $\mathbf{P}^l$. Therefore, in total there are $\prod_{l=1}^{L-1} N_l!$ equivalent networks.

Additionally, we can write

$$
\begin{aligned}
(\mathbf{P}^l\mathbf{W}^l)_{\text{new}} &= \mathbf{P}^l\mathbf{W}^l - \alpha\mathbf{P}^l\nabla_{\mathbf{W}^l}\mathcal{L} \\
&= \mathbf{P}^l\mathbf{W}^l - \alpha\mathbf{P}^l\delta^l(\mathbf{a}^{l-1})^{\mathrm{T}} \\
&= \mathbf{P}^l\mathbf{W}^l - \alpha\mathbf{P}^l \left[(\mathbf{W}^{l+1})^{\mathrm{T}}\delta^{l+1} \odot \sigma'(\mathbf{n}^l)\right] (\mathbf{a}^{l-1})^{\mathrm{T}} \\
&= \mathbf{P}^l\mathbf{W}^l - \alpha \left[(\mathbf{W}^{l+1}\mathbf{P}^{\mathrm{T}})^{\mathrm{T}}\delta^{l+1} \odot \sigma'(\mathbf{P}^l\mathbf{n}^l)\right] (\mathbf{a}^{l-1})^{\mathrm{T}} \\
&= \mathbf{P}^l\mathbf{W}^l - \alpha[(\mathbf{W}^{l+1}(\mathbf{P}^l)^{\mathrm{T}})^{\mathrm{T}}\delta^{l+1} \odot \sigma'(\mathbf{P}^l\mathbf{W}^l\mathbf{a}^{l-1} + \mathbf{P}^l\mathbf{b}^l)](\mathbf{a}^{l-1})^{\mathrm{T}}.
\end{aligned}
\tag{10}
$$

If we apply a permutation $\mathbf{P}^l$ to our update rule (equation 6) at any layer except the last, then using the above, we can express the gradient based update as

$$(\hat{\mathbf{W}}^l)_{\text{new}} = \hat{\mathbf{W}}^l - \alpha \left[(\hat{\mathbf{W}}^{l+1})^{\mathrm{T}}\delta^{l+1} \odot \sigma'(\hat{\mathbf{W}}^l\mathbf{a}^{l-1} + \hat{\mathbf{b}}^l)\right] (\mathbf{a}^{l-1})^{\mathrm{T}}\Box \tag{11}$$

The above implies that the permutations not only preserve the structural flow of information in the forward pass, but also preserve the structural flow of information during the update with the backward pass. That is we preserve the structural flow of information about the gradients with respect to the parameters during the backward pass trough the feed-forward layers.

**Permutation Equivalence for Convolutional Layers** We can easily extend the permutation equivalence in the feed-forward layers to convolution layers. Consider the 2D-convolution with input channel $\mathbf{x}$ and $O$ output channels $\mathbf{a}_{s_1} = \begin{bmatrix}\mathbf{a}_1 \\ \cdot \\ \mathbf{a}_O\end{bmatrix}$. We express a single convolution operation for the input channel $\mathbf{x}$ with a convolutional kernel $\mathbf{K}_o, o \in \{1, ..., O\}$ as

$$\mathbf{a}_o = \mathbf{b}_o + \mathbf{K}_o \star \mathbf{x}, o \in \{1, \cdots, O\}, \tag{12}$$

where $\star$ denotes the discrete convolution operation.

Note that in contrast to the hole set of permutation matrices $\mathbf{P}^l$ (which where introduced earlier) now we consider only a subset that affects the order of the input channels (if we have multiple) and the order of the concatenation of the output channels.

We now show that changing the order of input channels does not affect the output if the order of kernels is changed accordingly.

The proof is similar with the permutation equivalence for feed-forward layer. The difference here is that we take into account only the change in the order of channels and kernels. In order to prove permutation equivalence here it suffices to show that we can represent the convolution of multiple input channels by multiple convolution kernels in an alternative form, that is as matrix vector operation.

To do so we fist show that we can express the convolution of one input channel with one kernel to its equivalent matrix vector product form. Formally, we have that

$$\mathbf{a}_o = \mathbf{b}_o + \mathbf{K}_o \star \mathbf{x} = \mathbf{b}_o + \mathbf{R}_o\mathbf{x}, \tag{13}$$

where $\mathbf{R}_o$ is the convolution matrix. We build the matrix $\mathbf{R}_o$ in this alternative form (13) for the convolution operation from the convolutional kernel $\mathbf{K}_o$. $\mathbf{R}_o$ has a special structure (if we have 1D convolution then it is known as a circulant convolution matrix), while the input channel $\mathbf{x}$ remains the same. The number of columns in $\mathbf{R}_o$ equals the dimension of the input channel, while the number of rows in $\mathbf{R}_o$ equals the dimension of the output channel. In each row of $\mathbf{R}_o$, we store the elements of the convolution kernel. That is we sparsely distribute the kernel elements such that the multiplication of one row $\mathbf{R}_{o,j}$ of $\mathbf{R}_o$ with the input channel $\mathbf{x}$ results in the convolution output $a_{o,j}$ for the corresponding position $j$ at the output channel $\mathbf{a}_o$.

The convolution of one input channels by multiple kernels can be expressed as a matrix vector operation. In that case, the matrix in the equivalent form for the convolution with multiple kernels over one input represents a block concatenated matrix, where each of the block matrices has the previously described special structure, *i.e.*,

$$\mathbf{a}_{s_1} = \begin{bmatrix} \mathbf{a}_1 \\ \cdot \\ \mathbf{a}_O \end{bmatrix} = \begin{bmatrix} \mathbf{b}_1 \\ \cdot \\ \mathbf{b}_O \end{bmatrix} + \begin{bmatrix} \mathbf{R}_1 \\ \cdot \\ \mathbf{R}_O \end{bmatrix} \mathbf{x} = \mathbf{b}_f + \mathbf{R}_f \mathbf{x}, \tag{14}$$

where $\mathbf{b}_f = \begin{bmatrix} \mathbf{b}_1 \\ \cdot \\ \mathbf{b}_O \end{bmatrix}$ and $\mathbf{R}_f = \begin{bmatrix} \mathbf{R}_1 \\ \cdot \\ \mathbf{R}_O \end{bmatrix}$.

In the same way the convolution of multiple input channels $\mathbf{x}_1, ..., \mathbf{x}_S$ by multiple kernels $\mathbf{K}_1, ..., \mathbf{K}_O$ can be expressed as a matrix vector operation. In that case, the matrix in the equivalent form for the convolution with multiple kernels over multiple inputs represents a block diagonal matrix, where each of the blocks in the block diagonal matrix has the previously described special structure, *i.e.*,

$$\begin{bmatrix} \mathbf{a}_{s_1} \\ \cdot \\ \mathbf{a}_{s_s} \end{bmatrix} = \begin{bmatrix} \mathbf{b}_f \\ \cdot \\ \mathbf{b}_f \end{bmatrix} + \begin{bmatrix} \mathbf{R}_f & \mathbf{0} & ... & \mathbf{0} \\ \cdot & \cdot & \cdot & \cdot \\ \mathbf{0} & ... & \mathbf{0} & \mathbf{R}_f \end{bmatrix} \begin{bmatrix} \mathbf{x}_1 \\ \cdot \\ \mathbf{x}_S \end{bmatrix} = \begin{bmatrix} \mathbf{b}_f \\ \cdot \\ \mathbf{b}_f \end{bmatrix} + \mathbf{R} \begin{bmatrix} \mathbf{x}_1 \\ \cdot \\ \mathbf{x}_S \end{bmatrix}, \tag{15}$$

where $\mathbf{R} = \begin{bmatrix} \mathbf{R}_f & \mathbf{0} & ... & \mathbf{0} \\ \cdot & \cdot & \cdot & \cdot \\ \mathbf{0} & ... & \mathbf{0} & \mathbf{R}_f \end{bmatrix}$, which we can also express as

$$\mathbf{a} = \mathbf{b} + \mathbf{R} \begin{bmatrix} \mathbf{x}_1 \\ \cdot \\ \mathbf{x}_S \end{bmatrix}, \tag{16}$$

where $\mathbf{a} = \begin{bmatrix} \mathbf{a}_{s_1} \\ \cdot \\ \mathbf{a}_{s_s} \end{bmatrix}$ and $\mathbf{b} = \begin{bmatrix} \mathbf{b}_f \\ \cdot \\ \mathbf{b}_f \end{bmatrix}$.

Note that the above equation has equivalent form with equation (7), therefore, the previous proof is valid for the update with respect to hole matrix $\mathbf{R}$. However, $\mathbf{R}$ has a special structure, therefore for the update of of each element in $\mathbf{R}$, we can use the chain rule, which results in

$$\frac{\partial f(\mathbf{R})}{\partial R_{ij}} = \sum_k \sum_l \frac{\partial f(\mathbf{R})}{\partial R_{kl}} \frac{\partial R_{kl}}{\partial R_{ij}} = Tr \left[ \left[ \frac{\partial f(\mathbf{R})}{\partial \mathbf{R}} \right]^T \frac{\partial \mathbf{R}}{\partial R_{ij}} \right]. \tag{17}$$

Replacing $f()$ by $\mathcal{L}$ in the above and using the update rule equation (6) gives us the update equation for $R_{ij}$. Using similar argumentation and derivation that leads to equation (11) concludes the proof for permutation equivalence for a convolutional layer $\square$

**Empirical Evaluation** We empirically confirm the permutation equivalence (see Figure 1). We begin by comparing accuracies of permuted models (Figure 1 left). To that end, we randomly initialize model $A$ and train it for 10 epochs. We pick one random permutation, and permute all epochs of model $A$. For the permuted version $Ap$ we compute the test accuracy for all epochs. The test accuracy of model $A$ and $Ap$ lie on top of each other, so the permutation equivalence holds for the forward pass. To test the backwards pass, we create model $B$ as a copy of $Ap$ at initialization, and train for 10 epochs. Again, train and test epochs of $A$ and $B$ lie on top of each other, which indicates that the equivalence empirically holds for the backwards pass, too.

To track how models develop in weight space, we compute the mutual $\ell_2$-distances between the vectorized weights (Figure 1 right). The distance between $A$ and $Ap$, as well as between $A$ and $B$ is high and identical. Therefore, model $A$ is far away from models $Ap$ and $B$. Further, the distance

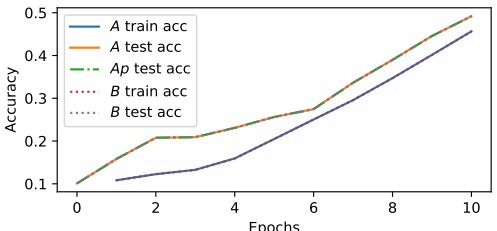 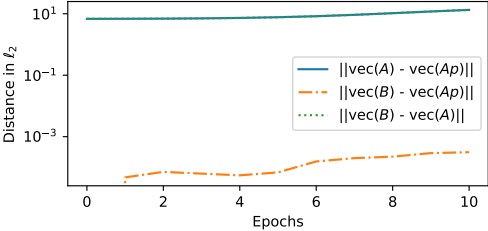

Figure 1: Empirical Evaluation of Permutation Equivalence. **Left:** Accuracies of orignal model $A$, permuted model $Ap$ and model $B$ trained from $Ap$'s initialization. All models are indistinguishable in their accuracies. **Right:** pairwise distances of vectorized weights over epochs of models $A$, $Ap$ and $B$. The distance between $A$ and $Ap$ as well as $A$ and $B$ is equally large and does not change much over the epochs. The distance between $Ap$ and $B$, which start from the same point in weight space, is small and remains small over the epochs. **both figures:** permuted versions of models are indistinguishable in their mapping, but far apart in weight space.

between $Ap$ and $B$ is small, confirming the backwards pass equivalence. We attribute the small difference to numerical errors.

**Further Weight Space Symmetries** It is important to note that besides the symmetry used above, other symmetries exist in the model weight space, which change the representation of a NN model, but not it's mapping, *e.g.*, scaling of subsequent layers with piece-wise linear activation functions [Dinh et al., 2017]. While some of these symmetries may be used as augmentation, these particular mappings only create equivalent networks in the forward pass, but different gradients and updates in the backward pass when back propagating. Therefore, we did not consider them in our work.

# B. Downstream Tasks Additional Details

In this appendix section, we provide additional details about the downstream tasks which we use to evaluate the utility of the hyper-representations obtained by our self-supervised learning approach.

## B.1 Downstream Tasks Problem Formulation

We use linear probing as a proxy to evaluate the utility of the learned hyper-representations, similar to Grill et al. [2020].

We denote the training and testing hyper-representations as $\mathbf{Z}_{train}$ and $\mathbf{Z}_{test}$. We assume that training $\mathbf{t}_{train}$ and testing $\mathbf{t}_{test}$ target vectors are given. We compute the closed form solution $\hat{\mathbf{r}}$ to the regression problem

$$(Q1) : \hat{\mathbf{r}} = \arg \min_{\mathbf{r}} \|\mathbf{Z}_{train}\mathbf{r} - \mathbf{t}_{train}\|_2^2,$$

and evaluate the utility of $\mathbf{Z}_{test}$ by measuring the $R^2$ score Wright [1921] as discrepancy between the predicted $\mathbf{Z}_{test}\hat{\mathbf{r}}$ and the true test targets $\mathbf{t}_{test}$.

Note that by using different targets $\mathbf{t}_{train}$ in $(Q1)$, we can estimate different $\mathbf{r}$ coefficients. This enables us to evaluate on different downstream tasks, including, accuracy prediction (Acc), epoch prediction (Eph) as proxy to model versioning, F-Score Goodfellow et al. [2016] prediction ($F_c$), learning rate (LR), $\ell_2$-regularization ($\ell_2$-reg), dropout (Drop) and training data fraction (TF). For these target values, we solve $(Q1)$, but for categorical hyper-parameters prediction, like the activation function (Act), optimizer (Opt), initialization method (Init), we train a linear perceptron by minimizing a cross entropy loss Goodfellow et al. [2016]. Here, instead of $R^2$ score, we measure the prediction accuracy.

## B.2 Downstream Tasks Targets

In this appendix subsection, we give the details about how we build the target vectors in the respective problem formulations for all of the downstream tasks.

**Accuracy Prediction (Acc).** In the accuracy prediction problem, we assume that for each trained NN model on a particular data set, we have its accuracy. Regarding the task of accuracy prediction, the value $a_{train,i}$ for the training NN models represents the training target value $t_{train,i} = a_{train,i}$, while the value $a_{test,j}$ for the testing NN models represents the true testing target value $t_{test,j} = a_{test,j}$.

**Generalization Gap Prediction (GGap).** In the generalization gap prediction problem, we assume that for each trained NN model on a particular data set, we have its train and test accuracy. The generalization gap represents the target value $g_i = a_{train,i} - a_{test,i}$ for the training NN models, while $g_j = a_{train,j} - a_{test,j}$ is the true target $t_{test,j} = g_j$ for the testing NN models.

**Epoch Prediction (Eph).** We consider the simplest setup as a proxy to model versioning, where we try to distinguish between NN weights and biases recorded at different epoch numbers during the training of the NNs in the model zoo. To that end, we assume that we construct the model zoo such that during the training of a NN model, we record its different evolving versions, *i.e.*, the zoo includes versions of one NN model at different epoch numbers $e_i$. Similarly to the previous task, our targets for the task of epoch prediction are the actual epoch numbers, $t_{train,i} = e_{train,i}$ and $t_{test,j} = e_{test,j}$, respectively.

**F Score Prediction** ($F_c$). To identify more fine-grained model properties, we therefore consider the class-wise F score. We define the F score prediction task similarly as in the previous downstream task. We assume that for each NN model in the training and testing subset of the model zoo, we have computed F score Goodfellow et al. [2016] for the corresponding class with label $c$ that we denote as $F_{train,c,i}$ and $F_{test,c,j}$, respectively. Then we use $F_{train,c,i}$ and $F_{test,c,i}$ as a target value $t_{train,c,i} = F_{train,c,i}$ in the regression problem and set $t_{test,c,j} = F_{test,c,j}$ during the test evaluation.

**Hyper-parameters Prediction**. We define the hyper-parameter prediction task identically as the previous downstream tasks. Where for continuous hyper-parameters, like learning rate (LR), $\ell_2$-regularization ($\ell_2$-reg), dropout (Drop), nonlinear thresholding function (TF), we solve the linear regression problem $(Q1)$. Similarly to the previous task, our targets for the task of hyper-parameters prediction are the actual hyper-parameters values.

In particular, for learning rate $t_{train,i} = learning\ rate$, for $\ell_2$-regularization $t_{train,i} = \ell_2\text{-}regularization\ type$, for dropout (Drop) $t_{train,i} = dropout\ value$ and for nonlinear thresholding function (TF) $t_{train,i} = nonlinear\ thresholding\ function$. In a similar fashion, we also define the test targets $\mathbf{t}_{test}$.

For categorical hyper-parameters, like activation function (Act), optimizer (Opt), initialization method (Init), instead of regression loss $(Q1)$, we train a linear perception by minimizing a cross entropy loss Goodfellow et al. [2016]. Here, also we also define the targets as detailed above. The only difference here is that the targets here have discrete categorical values.

| Our Zoos | Data | NN Type | No. Param. | Varying Prop. | No. Eph | No. NNs |
|---|---|---|---|---|---|---|
| TETRIS-SEED | TETRIS | MLP | 100 | SEED (1-1000) | 75 | 1000*75 |
| TETRIS-HYP | TETRIS | MLP | 100 | SEED (1-100), ACT, INIT, LR | 75 | 2900*75 |
| MNIST-SEED | MNIST | CNN | 2464 | SEED (1-1000) | 25 | 1000*25 |
| FASHION-SEED | F-MNIST | CNN | 2464 | SEED (1-1000) | 25 | 1000*25 |
| MNIST-HYP-1-FIX-SEED | MNIST | CNN | 2464 | FIXED SEED, ACT, INIT, LR | 25 | $\sim 1152{*}25$ |
| MNIST-HYP-1-RAND-SEED | MNIST | CNN | 2464 | RANDOM SEED, ACT, INIT, LR | 25 | $\sim 1152{*}25$ |
| MNIST-HYP-5-FIX-SEED | MNIST | CNN | 2464 | 5 FIXED SEEDS, ACT, INIT, LR | 25 | $\sim 1280{*}25$ |
| MNIST-HYP-5-RAND-SEED | MNIST | CNN | 2464 | 5 RANDOM SEEDS, ACT, INIT, LR | 25 | $\sim 1280{*}25$ |

| Existing Zoos | Data | NN Type | No. Param. | Varying Prop. | No. Eph | No. NNs |
|---|---|---|---|---|---|---|
| MNIST-HYP | MNIST | CNN | 4970 | ACT, INIT, OPT, LR, $\ell_2$-REG, DROP, TF | 9 | $\sim 30000{*}9$ |
| FASHION-HYP | F-MNIST | CNN | 4970 | ACT, INIT, OPT, LR, $\ell_2$-REG, DROP, TF | 9 | $\sim 30000{*}9$ |
| CIFAR10-HYP | CIFAR10 | CNN | 4970 | ACT, INIT, OPT, LR, $\ell_2$-REG, DROP, TF | 9 | $\sim 30000{*}9$ |
| SVHN-HYP | SVHN | CNN | 4970 | ACT, INIT, OPT, LR, $\ell_2$-REG, DROP, TF | 9 | $\sim 30000{*}9$ |

| | 1 Init | $M$ Init | No data leakage | Dense check points |
|---|---|---|---|---|
| Unterthiner et al. [2020] | $\checkmark$ | $\checkmark$ | $\times$ | $\times$ |
| Eilertsen et al. [2020] | $\checkmark$ | $\times$ | $\checkmark$ | $\times$ |
| proposed zoos | $\checkmark$ | $\checkmark$ | $\checkmark$ | $\checkmark$ |

Table 1: Overview of the characteristics for the model zoos proposed and used (existing) in this work.

## C. Model Zoos Details

In Table 1, we give an overview of the characteristics for the used model zoos in this paper. This includes

- The data sets used for zoo creation.

- The type of the NN models in the zoo.

- Number of learnable parameters for each of the NNs.

- Used number of model versions that are taken at the corresponding epochs during training.

- Total number of NN models contained in the zoo.

In Table 1 we also compare the existing and the introduced model zoos in prior and this work in terms of properties like initialization, data leakage and presence of dense model versions obtained by recording the NN model during training evolution.

In Table 2 we provide the architecture configurations and exact modes of variation of our model zoos.

| OUR ZOOS | INIT | SEED | OPT | ACT | LR | DROP | $\ell_2$-REG |
|---|---|---|---|---|---|---|---|
| TETRIS-SEED | UNIFORM | 1-1000 | ADAM | TANH | 3E-5 | 0.0 | 0.0 |
| TETRIS-HYP | UNIFORM, NORMAL, KAIMING-NO, KAIMING-UN, XAVIER-NO, XAVIER-UN, | 1-100 | ADAM | TANH, RELU | 1E-3, 1E-4, 1E-5 | 0.0 | 0.0 |
| MNIST-SEED | UNIFORM | 1-1000 | ADAM | TANH | 3E-4 | 0.0 | 0.0 |
| MNIST-HYP-1-FIX-SEED | UNIFORM, NORMAL, KAIMING-UN, KAIMING-NO | 42 | ADAM, SGD | TANH, RELU, SIGMOID, GELU | 3E-3, 1E-3, 3E-4, 1E-4 | 0.0, 0.3, 0.5 | 0, 1E-3, 1E-1 |
| MNIST-HYP-1-RAND-SEED | UNIFORM, NORMAL, KAIMING-UN, KAIMING-NO | $1 \in [1e0, 1e6]$ | ADAM, SGD | TANH, RELU, SIGMOID, GELU | 3E-3, 1E-3, 3E-4, 1E-4 | 0.0, 0.3, 0.5 | 0, 1E-3, 1E-1 |
| MNIST-HYP-5-FIX-SEED | UNIFORM, NORMAL, KAIMING-UN, KAIMING-NO | 1,2,3,4,5 | ADAM, SGD | TANH, RELU, SIGMOID, GELU | 1E-3, 1E-4 | 0.0, 0.5 | 1E-3, 1E-1 |
| MNIST-HYP-5-RAND-SEED | UNIFORM, NORMAL, KAIMING-UN, KAIMING-NO | $5 \in [1e0, 1e6]$ | ADAM, SGD | TANH, RELU, SIGMOID, GELU | 1E-3, 1E-4 | 0.0, 0.5 | 1E-3, 1E-1 |
| FASHION-SEED | UNIFORM | 1-1000 | ADAM | TANH | 3E-4 | 0.0 | 0.0 |

Table 2: Architecture configurations and modes of variation of our model zoos.

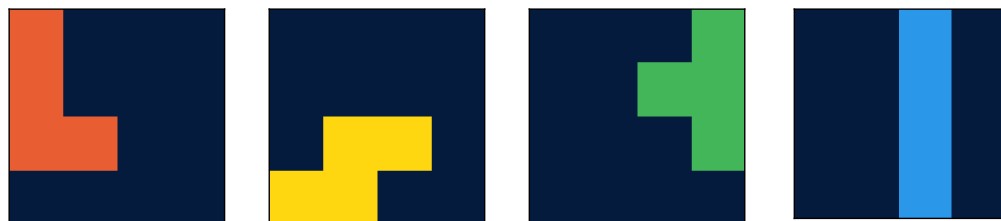

Figure 2: Visualization of samples representing the four basic shapes in our Tetris data set.

## C.1 Zoos Generation Using Tetris Data

As a toy example, we first create a 4x4 grey-scaled image data set that we call *Tetris* by using four tetris shapes. In Figure 2 we illustrate the basic shapes of the tetris data set. We introduce two zoos, which we call TETRIS-SEED and TETRIS-HYP, which we group under *small*. Both zoos contain FFN with two layers. In particular, the FFN has input dimension of $16$ a latent dimension of $5$ and output dimension of $4$. In total the FFN has $16 \times 5 + 5 \times 4 = 100$ learnable parameters (see Table ). We give an illustration of the used FFN architecture in Figure 5.

In the TETRIS-SEED zoo, we fix all hyper-parameters and vary only the seed to cover a broad range of the weight space. The TETRIS-SEED zoo contains 1000 models that are trained for 75 epochs. In total, this zoo contains $1000 \times 75 = 75000$ trained NN weights and biases.

To enrich the diversity of the models, the TETRIS-HYP zoo contains FFNs, which vary in activation function [tanh, relu], the initialization method [uniform, normal, kaiming normal, kaiming uniform, xavier normal, xavier uniform] and the learning rate [$1e$-3, $1e$-4, $1e$-5]. In addition,

each combination is trained with 100 different seeds. Out of the 3600 models in total, we have successfully trained 2900 for 75 epochs - the remainders crashed and are disregarded. So in total, this zoo contains $2900 \times 75 = 217500$ trained NN weights and biases.

## C.2 Zoos Generation Using MNIST Data

Similarly to `TETRIS-SEED`, we further create medium sized zoos of CNN models. In total the CNN has 2464 learnable parameters, distributed over 3 convolutional and 2 fully connected layers. The full architecture is detailed in Table . We give an illustration of the used CNN architecture in Figure 6. Using the MNIST data set, we created five zoos with approximately the same number of CNN models.

In the `MNIST-SEED` zoo we vary only the random seed (1-1000), while using only one fixed hyper-parameter configuration. In particular,

In `MNIST-HYP-1-FIX-SEED` we vary the hyper-parameters. We use only one fixed seed for all the hyper-parameter configurations (similarly to [Unterthiner et al., 2020]). The `MNIST-HYP-1-RAND-SEED` model zoo contains CNN models, where per each model we draw and use 1 random seeds and different hyper-parameter configuration.

We generate `MNIST-HYP-5-FIX-SEED` insuring that for each hyper-parameter configurations we add 5 models that share 5 fixed seed. We build `MNIST-HYP-5-RAND-SEED` such that for each hyper-parameter configurations we add 5 models that have different random seeds.

We grouped these model zoos as *medium*. In total, each of these zoos approximately $1000 \times 25 = 25000$ trained NN weights and biases.

In Figure 3 we provide a visualization for different properties of the `MNIST-SEED`, `MNIST-HYP-1-FIX-SEED`, `MNIST-HYP-1-RANDOM-SEED`, `MNIST-HYP-5-FIX-SEED` and `MNIST-HYP-5-RANDOM-SEED` zoos. The visualization supports the empirical findings from the paper, that zoos which vary in seed only appear to contain a strong correlation between the mean of the weights and the accuracy. In contrast, the same correlation is considerably lower if the hyper-parameters are varied. Further, we also observe clusters of models with shared initialization method and activation function for zoos with fixed seeds. Random seeds seem to disperse these clusters to some degree. This additionally confirms our hypothesis about the importance of the generating factors for the zoos. We find that zoos containing hyper-parameters variation and multiple (random) seeds, have a rich set of properties, avoid 'shortcuts' between the weights (or their statistics) and properties, and therefore benefits hyper-representation learning.

In Figure 4, we show additional UMAP reductions of `MNIST-HYP`, which confirm our previous findings. Similarly to the UMAP for the `MNIST-HYP-1-FIX-SEED` zoo, the UMAP for the `MNIST-HYP` has distinctive and recognizable initialization points. The categorical hyper-parameters are visually separable in weight space. As we can see in the same figure, it seems that the UMAP for the `MNIST-HYP` zoo contains very few paths along which the evolution during learning of all the models can be tracked in weight space, facilitating both epoch and accuracy prediction.

## C.3 Zoo Generation Using F-MNIST Data

We used the F-MNIST data set. As for the previous zoos for the MNIST data set, we have created one zoos with exactly the same number of CNN models as in `MNIST-SEED`. In this zoo that we call `FASHION-SEED`, we vary only the random seed (1-1000), while using only one fixed hyper-parameter configuration.

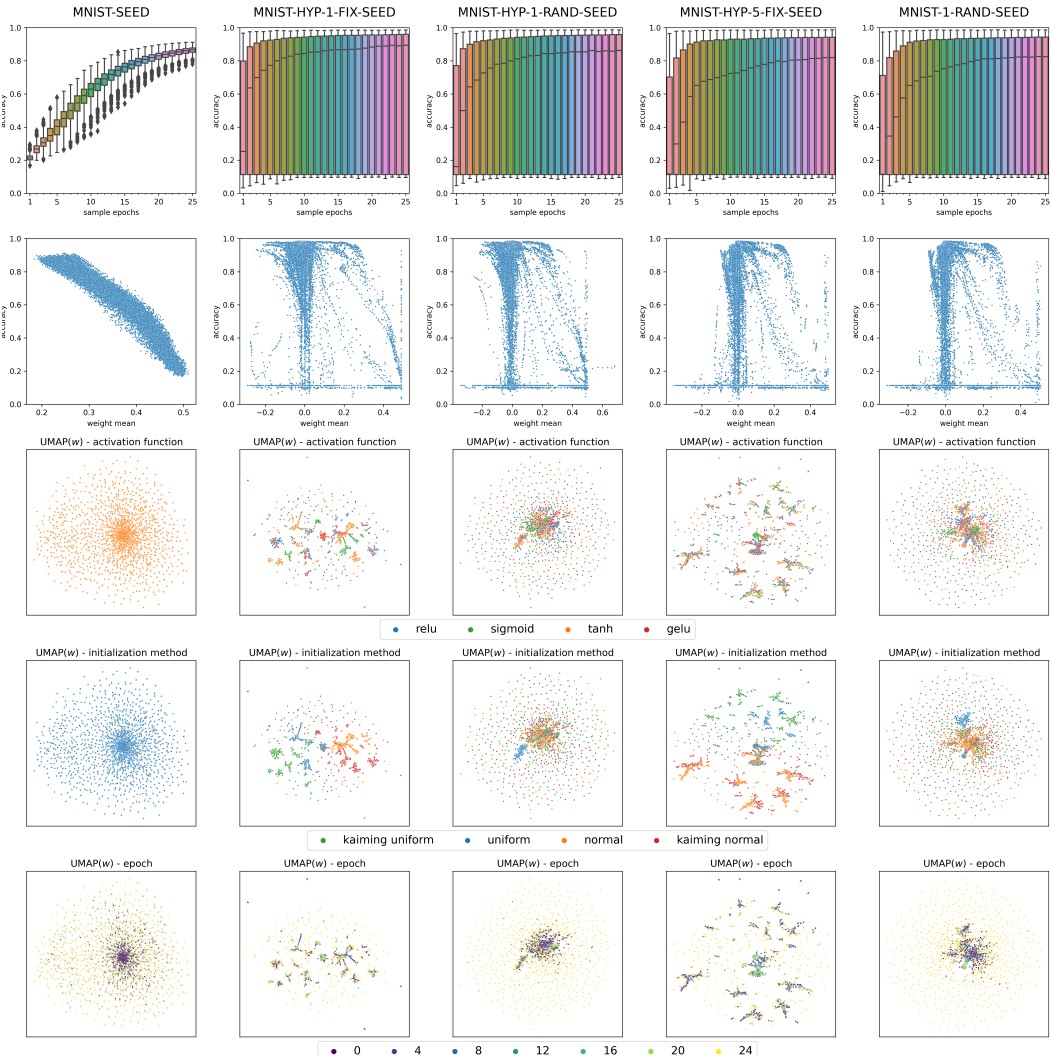

Figure 3: Visualization on the properties for the `MNIST-SEED`, `MNIST-HYP-1-FIX-SEED`, `MNIST-HYP-1-RANDOM-SEED`, `MNIST-HYP-5-FIX-SEED` and `MNIST-HYP-5-RANDOM-SEED` zoos. **Row One**. Boxplot of NNs accuracy over the epoch ids. **Row Two**. NNs accuracy plotted over the mean of the NNs weights of each sample. `MNIST-SEED` shows homogeneous development and a strong correlation between weight mean and accuracy, while varying the hyperparameters yields heterogeneous development without that correlation. **Rows Three to Five**. UMAP reductions of the weight space coloured by activation function, initialization method and sample epoch. Zoos with fixed seeds contain visible clusters of NNs that share same initialization method or activation function. Zoos with varying hyperparameters and random seeds do not contain such clear clusters.

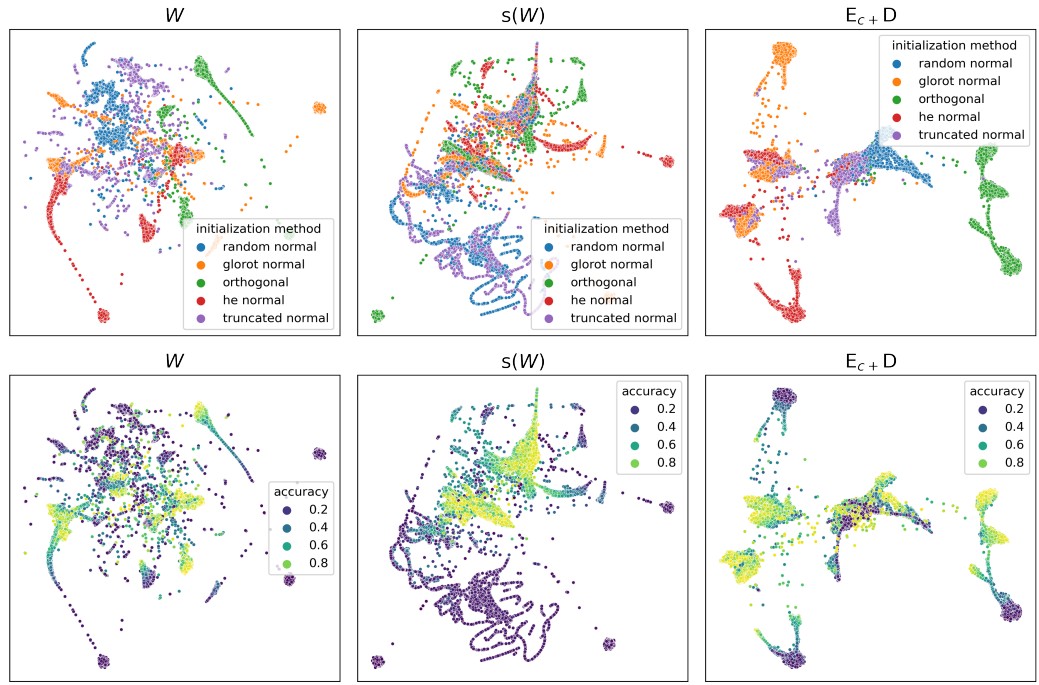

Figure 4: UMAP dimensionality reduction of the weight space (left), weight statistics (middle) and learned hyper-representations (right) for the MNIST-HYP zoo Unterthiner et al. [2020]. The initialization methods for the trained NN weights are already visually separable to a high degree in weight space, which carries over to the learned embedding space, while the statistics introduce a mix between the initialization methods. For accuracy, in seems that the statistics filter out and contain more relevant information than the weight space. Learned embeddings appears to cluster the models according to their initialization methods and within the clusters help to preserve high accuracy.

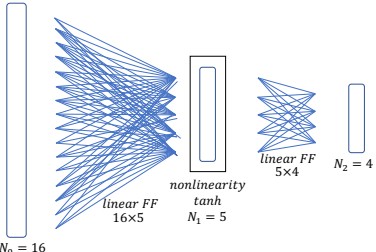

Figure 5: A diagram for the feed-forward architecture of the NNs in the `TETRIS-SEED` and `TETRIS-HYP` zoos.

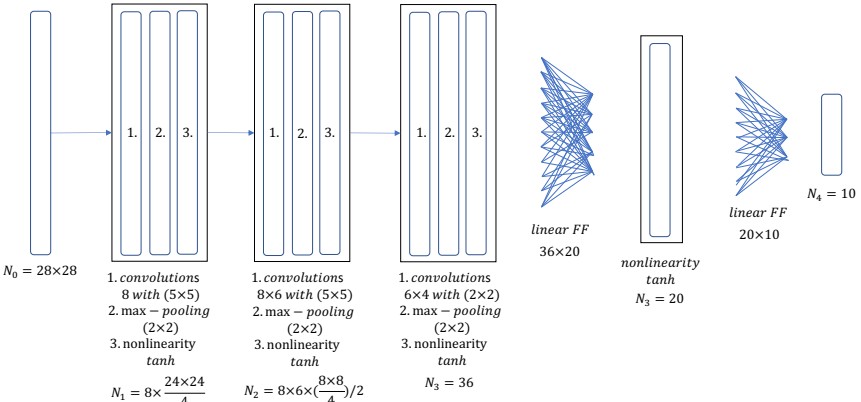

Figure 6: A diagram for the CNN architecture of the NNs in the MNIST zoos.

| | TYPE | DETAILS | PARAMS |
|---|---|---|---|
| 1.1 | LINEAR | CH-IN=16, CH-OUT=5 | 80 |
| 1.2 | NONLIN. | TANH | |
| 2.1 | LINEAR | CH-IN=5, CH-OUT=4 | 20 |

Table 3: FFN Architecture Details. CH-IN describes the number of input channels, CH-OUT the number of output channels.

| | TYPE | DETAILS | PARAMS |
|---|---|---|---|
| 1.1 | CONV | CH-IN=1, CH-OUT=8, KS=5 | 208 |
| 1.2 | MAXPOOL | KS= 2 | |
| 1.3 | NONLIN. | TANH | |
| 2.1 | CONV | CH-IN=8, CH-OUT=6, KS=5 | 1206 |
| 2.2 | MAXPOOL | KS= 2 | |
| 2.3 | NONLIN. | TANH | |
| 3.1 | CONV | CH-IN=6, CH-OUT=4, KS=2 | 100 |
| 3.2 | MAXPOOL | KS= 2 | |
| 3.3 | NONLIN. | TANH | |
| 4 | FLATTEN | | |
| 5.1 | LINEAR | CH-IN=36, CH-OUT=20 | 740 |
| 5.2 | NONLIN. | TANH | |
| 6.1 | LINEAR | CH-IN=20, CH-OUT=10 | 200 |

Table 4: CNN Architecture Details. CH-IN describes the number of input channels, CH-OUT the number of output channels. KS denotes the kernel size, kernels are always square.

## D. Additional Results

In this appendix section, we provide additional results about the impact of the compression ratio $c = N/L$.

### D.1 Impact of the Compression Ratio N/L

In this subsection, we first explain the experiment setup and then comment on the results about the impact of the compression ratio on the performance for downstream tasks.

**Experiment Setup**. To see the impact of the compression ratio $c = N/L$ on the performance over the downstream tasks, we use our hyper-representation learning approach under different types of architectures, including $E_c$, ED and $E_cD$ (see Section 3 in the paper). As encoders E and decoders D, we used the attention-base modules introduced in Section 3 in the paper. The attention-based encoder and decoder, on the TETRIS-SEED and TETRIS-HYP zoos, we used 2 attention blocks with 1 attention head each, token dimensions of 128 and FC layers in the attention module of dimension 512.

We use our weight augmentation methods for representation learning (please see Section 3.1 in the paper). We run our representation learning algorithm for up to 2500 epochs, using the adam optimizer Kingma and Ba [2014], a learning rate of 1e-4, weight decay of 1e-9, dropout of 0.1 percent and batch-sizes of 500. In all of our experiments, we use 70% of the model zoos for training and 15% for validation and testing each. We use checkpoints of all epochs, but ensure that samples from the same models are either in the train or in the test split of the zoo. As quality metric for the self-supervised learning, we track the reconstruction $R^2$ on the test split of the zoo.

**Results**. As Table 5 shows, all NN architectures decrease in performance, as the compression ratio increases. The purely contrastive setup $E_c$ generally learns embeddings which are useful for the downstream tasks, which are very stable under compression. These results strongly depend on a projection head with enough complexity. The closer the contrastive loss comes to the bottleneck of the encoder, the stronger the downstream tasks suffer under compression. Notably, the reconstruction of ED is very stable, even under high compression ratios. However, higher compression ratios appear to negatively impact the hyper-representations for the downstream tasks we consider here. The combination of reconstruction and contrastive loss shows the best performance for $c = 2$, but suffers under compression. Higher compression ratios perform comparably on the downstream tasks, but don't manage high reconstruction $R^2$. We interpret this as sign that the combination of losses requires high capacity bottlenecks. If the capacity is insufficient, the two objectives can't be both satisfied.

### D.2 NN Model Characteristics Prediction on FASHION-SEED

Due to space limitations, here in Figure 6, we present the results on the FASHION-SEED together with the results on MNIST-SEED. The experimental setup is same as for the MNIST-SEED zoo, which is explained in the paper. Here, we add a complementary result to our ablation study about the seed variation, that we presented in section 4.3 in the paper. Similarly to the discussion in the paper, random seeds variation in the FASHION-SEED again appears to make the prediction more challenging. The results show that the proposed approach is on-par with the comparing $s(W)$ for this type of model zoos.

### D.3 In-distribution and Out-of-distribution Prediction

In Figures 7, 8 and 9 we show in-distribution and out-of-distribution comparative results for test accuracy, epoch id and generalization gap prediction using the MNIST-HYP zoo.

In the majority of the results for accuracy and generalization gap prediction, our learned representations have higher $R^2$ and Kendall's $\tau$ score. Also, the baseline methods the distribution of predicted target values is more dispersed compared to the true target values. On the epoch id prediction we have comparable results but with lower score, we attribute this to the fact that the zoos contain sparse check points and we suspect that there are not enough so that our learning model could capture the present variability. Overall in the in-distribution and out-of-distribution results for test accuracy, epoch id and generalization gap prediction, the proposed approach has a slight advantage.

Encoder with contrastive loss $E_c$

| $c$ | REC | EPH | ACC | GGAP | $F_{C0}$ | $F_{C1}$ | $F_{C2}$ | $F_{C3}$ |
|---|---|---|---|---|---|---|---|---|
| 2 | – | 96.7 | 90.8 | 82.5 | 67.7 | 72.0 | 74.4 | 85.8 |
| 3 | – | 96.6 | 89.4 | 81.5 | 68.4 | 69.4 | 71.1 | 85.1 |
| 5 | – | 96.4 | 89.5 | 81.8 | 67.1 | 68.7 | 69.7 | 84.0 |

Encoder and decoder with reconstruction loss ED

| $c$ | REC | EPH | ACC | GGAP | $F_{C0}$ | $F_{C1}$ | $F_{C2}$ | $F_{C3}$ |
|---|---|---|---|---|---|---|---|---|
| 2 | 96.1 | 88.3 | 68.9 | 69.9 | 47.8 | 57.2 | 33.0 | 58.1 |
| 3 | 93.0 | 74.6 | 69.4 | 66.9 | 53.5 | 46.5 | 38.9 | 48.3 |
| 5 | 87.7 | 80.5 | 60.0 | 63.3 | 37.9 | 48.8 | 24.4 | 52.6 |

Encoder and decoder with reconstruction and contrastive loss $E_c$D

| $c$ | REC | EPH | ACC | GGAP | $F_{C0}$ | $F_{C1}$ | $F_{C2}$ | $F_{C3}$ |
|---|---|---|---|---|---|---|---|---|
| 2 | 84.1 | 97.0 | 90.2 | 81.9 | 70.7 | 75.9 | 69.4 | 86.6 |
| 3 | 75.6 | 96.3 | 88.3 | 80.7 | 66.9 | 70.8 | 66.1 | 83.2 |
| 5 | 64.5 | 96.3 | 85.2 | 80.0 | 63.5 | 68.0 | 61.3 | 73.6 |

Table 5: The impact of the compression ratio $c = N/L$ in the different NN architectures of our approach for learning hyper-representations over the `Tetris-Seed` Model Zoo. All values are $R^2$ scores and given in %.

| | MNIST-SEED | | | FASHION-SEED | | |
|---|---|---|---|---|---|---|
| | W | s(W) | $E_{c+}$D | W | s(W) | $E_{c+}$D |
| EPH | 84.5 | **97.7** | 97.3 | 87.5 | **97.0** | 95.8 |
| ACC | 91.3 | 98.7 | **98.9** | 88.5 | 97.9 | **98.0** |
| GGAP | 56.9 | 66.2 | **66.7** | 70.4 | 81.4 | **83.2** |

Table 6: $R^2$ score in % for epoch, accuracy and generalization gap.

Due to space limitations, for the `MNIST-SEED`, `FASHION-SEED` zoos and an additional `SVHN-SEED` zoo we only include out-of-distribution results for accuracy prediction in Figure 10. Here, too, our learned representations have higher scores in both Kendall's $\tau$ as well as $R^2$. Further, the accuracy prediction for `SVHN-SEED` clearly preserves the order, but has a noticable bias. We attribute that effect to the different accuracy distributions of `MNIST-SEED` (ID, accuracy: [0.2,0.95]) and `SVHN-SEED` (OOD, accuracy: [0.2,0.75]). Due to the higher accuracy in `MNIST-SEED`, we suspect that the accuracy in `SVHN-SEED` is overestimated.

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

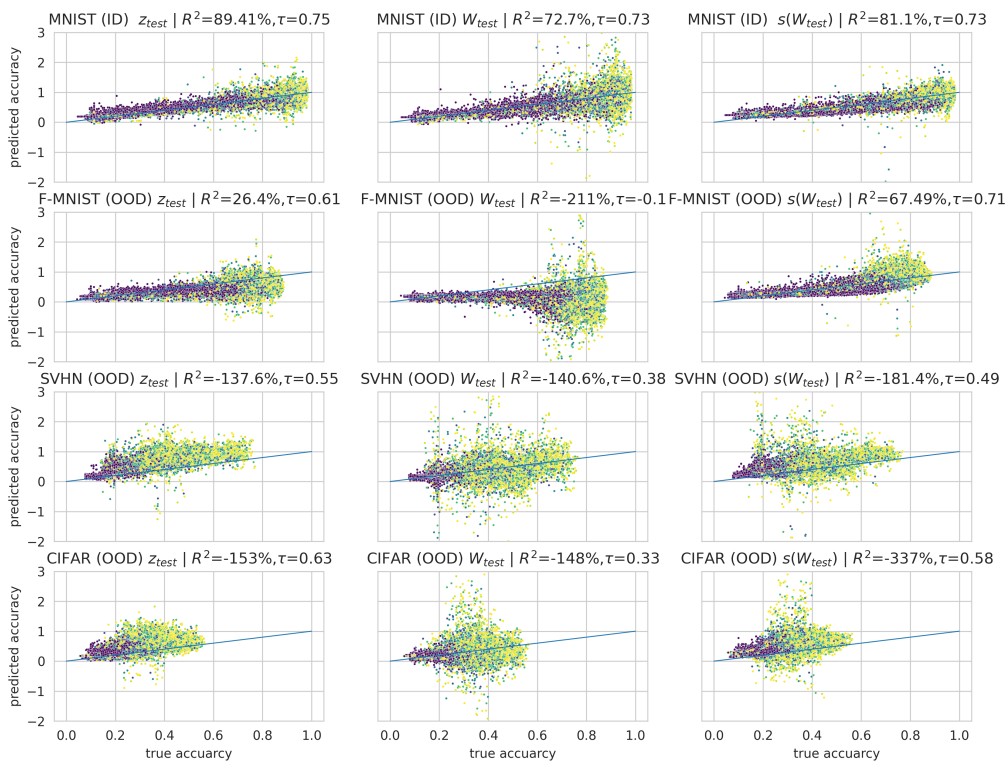

| | MNIST-HYP | | | FASHION-HYP | | | SVHN-HYP | | | CIFAR10-HYP | | |
|---|---|---|---|---|---|---|---|---|---|---|---|---|
| | W | s(W) | $E_{c}$+D | W | s(W) | $E_{c}$+D | W | s(W) | $E_{c}$+D | W | s(W) | $E_{c}$+D |
| MNIST-HYP ($\tau$) | .73 | .73 | **.75** | -.08 | **.71** | .61 | .38 | .49 | **.55** | .33 | .58 | **.63** |
| MNIST-HYP ($R^2$) | 72.7 | 81.1 | **89.4** | -211 | **67** | 26 | -140 | -180 | **-137** | **-148** | -337 | -153 |

Figure 7: In-distribution and out-of-distribution results for test accuracy prediction. Representation learning model and linear probes are trained on `MNIST-HYP`, and evaluated on `MNIST-HYP`, `FASHION-HYP`, `SVHN-HYP` and `CIFAR-HYP`.

Diederik P. Kingma and Jimmy Ba. Adam: A method for stochastic optimization. *CoRR*, abs/1412.6980, 2014.

David E Rumelhart, Geoffrey E Hinton, and Ronald J Williams. Learning representations by back-propagating errors. *nature*, 323(6088):533–536, 1986. Publisher: Nature Publishing Group.

Thomas Unterthiner, Daniel Keysers, Sylvain Gelly, Olivier Bousquet, and Ilya Tolstikhin. Predicting Neural Network Accuracy from Weights. *arXiv:2002.11448 [cs, stat]*, February 2020. URL `http://arxiv.org/abs/2002.11448`. arXiv: 2002.11448.

Sewall Wright. Correlation and causation. *J. agric. Res.*, 20:557–580, 1921.

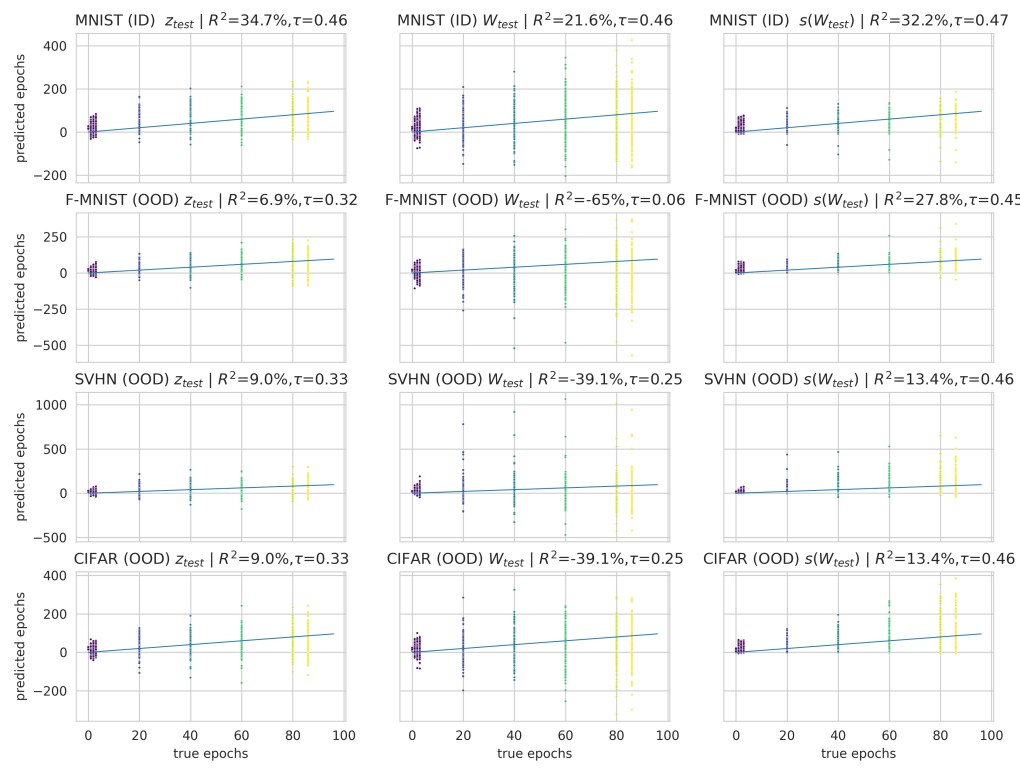

| | MNIST-HYP | | | FASHION-HYP | | | SVHN-HYP | | | CIFAR10-HYP | | |
|---|---|---|---|---|---|---|---|---|---|---|---|---|
| | W | s(W) | $E_{c+}$D | W | s(W) | $E_{c+}$D | W | s(W) | $E_{c+}$D | W | s(W) | $E_{c+}$D |
| MNIST-HYP ($\tau$) | .46 | **.47** | .46 | .06 | **.45** | .32 | .25 | **.46** | .33 | .18 | **.41** | .16 |
| MNIST-HYP ($R^2$) | 21.6 | 32.2 | **34.7** | -64.9 | **27.8** | 6.9 | -39.1 | **13.4** | 9. | -21.9 | **19.2** | -13. |

Figure 8: In-distribution and out-of-distribution results for the epoch id predictions. Representation learning model and linear probes are trained on MNIST-HYP, and evaluated on MNIST-HYP, FASHION-HYP, SVHN-HYP and CIFAR-HYP.

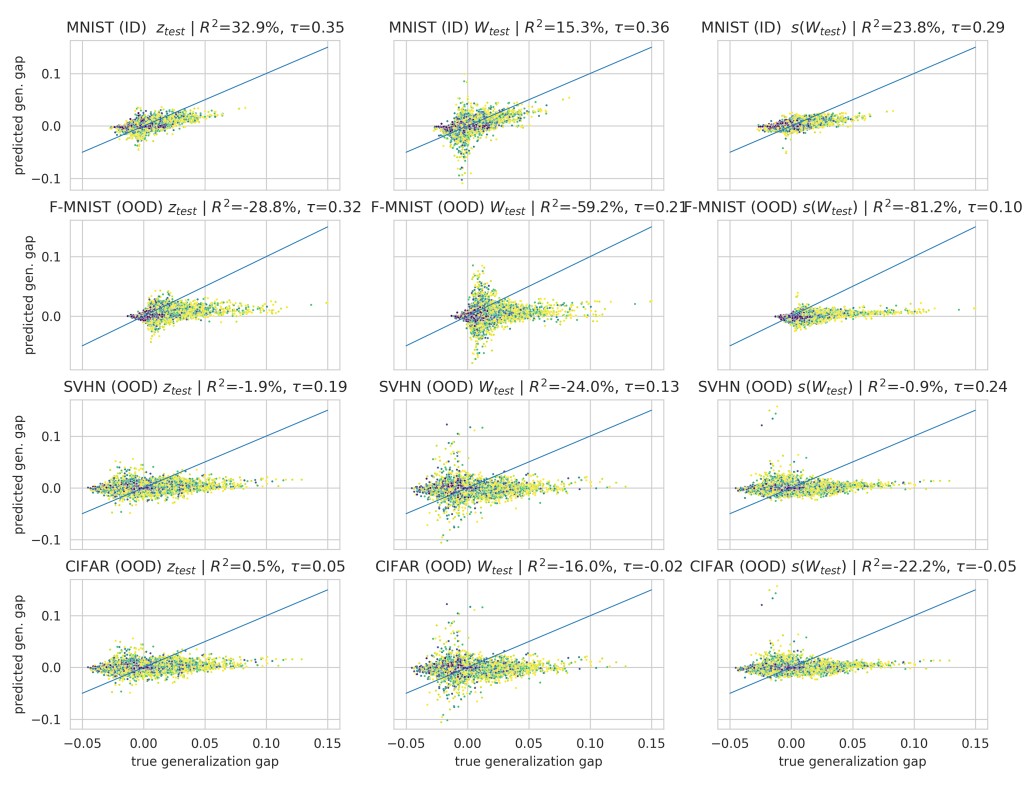

| | MNIST-HYP | | | FASHION-HYP | | | SVHN-HYP | | | CIFAR10-HYP | | |
|---|---|---|---|---|---|---|---|---|---|---|---|---|
| | W | s(W) | $E_{c+}$D | W | s(W) | $E_{c+}$D | W | s(W) | $E_{c+}$D | W | s(W) | $E_{c+}$D |
| MNIST-HYP ($\tau$) | **.36** | .29 | .35 | .20 | .10 | **.32** | .13 | **.24** | .19 | -.05 | -.02 | **.05** |
| MNIST-HYP ($R^2$) | 15.3 | 24.8 | **32.9** | -56.2 | -81.8 | **-27.8** | -24. | **-.9** | -1.9 | -16. | -22.2 | **.5** |

Figure 9: In distribution and out-of-distribution results for the generalization gap predictions. Representation learning model and linear probes are trained on MNIST-HYP, and evaluated on MNIST-HYP, FASHION-HYP, SVHN-HYP and CIFAR-HYP.

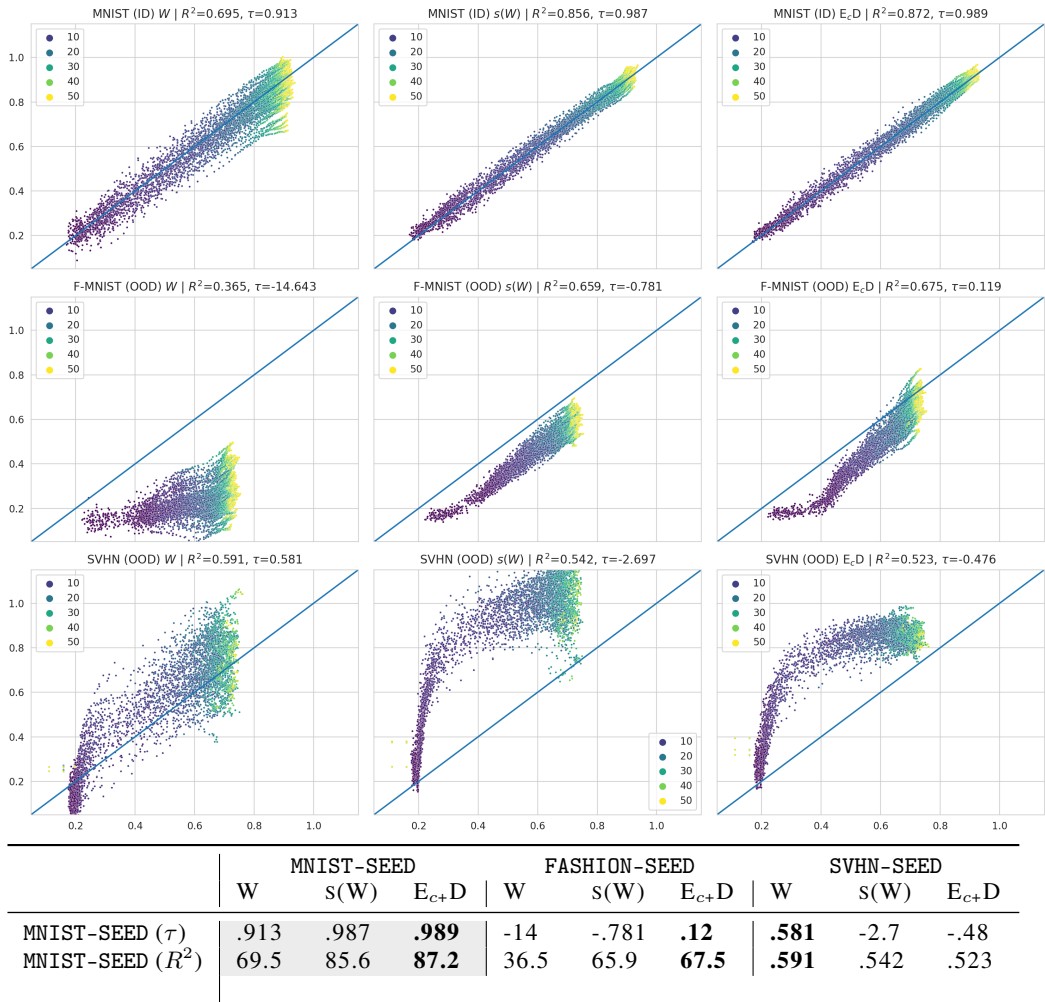

| | MNIST-SEED | | | FASHION-SEED | | | SVHN-SEED | | |
|---|---|---|---|---|---|---|---|---|---|
| | W | s(W) | $E_{c+}D$ | W | s(W) | $E_{c+}D$ | W | s(W) | $E_{c+}D$ |
| MNIST-SEED ($\tau$) | .913 | .987 | **.989** | -14 | -.781 | **.12** | **.581** | -2.7 | -.48 |
| MNIST-SEED ($R^2$) | 69.5 | 85.6 | **87.2** | 36.5 | 65.9 | **67.5** | **.591** | .542 | .523 |

Figure 10: In-distribution and out-of-distribution results for test accuracy prediction. Representation learning model and linear probes are trained on MNIST-SEED, and evaluated on MNIST-SEED, FASHION-SEED and SVHN-SEED.