# OpenReview forum: "Self-Supervised Representation Learning on Neural Network Weights for Model Characteristic Prediction"
_NeurIPS.cc/2021/Conference — NeurIPS 2021 Poster_

### Official Review · Reviewer_VBuX · 2021-07-14

**Rating:** 6
**Confidence:** 3

**Summary:**

The authors propose self-supervised algorithms to predict the model characteristics from the neural network weights. Authors experimented with different self-supervised objectives based on autoencoder, contrastive learning (NT_Xent), and modified contrastive loss. They utilize different architectures based on Fully-Connected Feed-Forward Networks (FF) and transformer (Att).

Extensive experiments show that neural representation learned with self-supervision recovers latent structure from the neural network's weights. The learned representations outperform approaches based on features from weights, different modifications of PCA, UMAP, and layer-wise weight statistics (Unterthiner, 2020) in multiple downstream tasks and out-of-distribution generalization.

Additionally motivated by the unresolved question: whether model zoo should contain multiple seeds or not due to possible leakage between samples, - they generate new model zoos based on MNIST and show that seed variation for each set of the hyperparameters is beneficial to learn better representation to recover NN's properties.

**Limitations And Societal Impact:**

The authors have adequately addressed the limitations and potential negative social impact of their work.

**Main Review:**

Strengths:

- This work is more about a toy benchmark for learning from the NN's weights and applying the self-supervision to the specific domain. The experiments and ablations are extensive.
- The experiments clarified the question about using seeds in model zoos.

Weaknesses:

- There is no supervised baseline, and I fail to see a motivation for self-supervision. The labels and data can be easily generated and synthesized. Most of the SOTA results are still achieved with Supervised models. One could take the encoder from your architectures then add linear layers to train for regression and classification in a multi-task fashion. Then one can take representation before linear classification layers and perform linear evaluation or visualization with UMAP.

- The experiments show that self-supervision works for this domain. However, I do not see them particularly useful for real case scenarios and insights:
  - It will be interesting to see a different evaluation. For example, the researcher has already trained many models. Then the researcher would use the proposed characteristic prediction model to evaluate new models. Finally, let's ask how close the prediction of the characteristic model will be to the actual performance of the fully converged model given the new weights at the initialization step or after a couple of epochs. The main goal is to save compute and time while trying to find a better model.
  - The experiments do not uncover new insights about how neural networks work.

Minor:

- Lines 298-299: " that changing only the seeds results in models with very similar evolution during learning," Could we consider seed just as a data augmentation?

- Line 84: It is unclear what the learning procedure for the model is and how it is different from hyper-parameters.

- Line 124: The equation for the modified contrastive loss, it seems, has an only part for positive pairs but not a negative part. I am not sure if it can be called contrastive, and it looks like a cosine similarity that has been used in at least two works as SimSiam [1] and SPR [2].
  - [1] Chen, Xinlei, and Kaiming He. "Exploring simple siamese representation learning." *Proceedings of the IEEE/CVF Conference on Computer Vision and Pattern Recognition*. 2021.
  - [2] Schwarzer, Max, et al. "Data-Efficient Reinforcement Learning with Self-Predictive Representations." *International Conference on Learning Representations*. 2020.
  - The equation has missing parathesis for the exponent.

- Line 216: Given the text, it seems that the model selection was performed on the test set, which is supposed to be a hold-out set but not as a validation set.

- Line 258: I do not purely agree that vanilla transformer does well with long-range relations, as the community is still trying to solve these questions (e.g., Transformer-XL [3], XLNet [4], Feedback Transformer [5]). The authors should clarify this.
  - [3] Dai, Zihang, et al. "Transformer-XL: Attentive Language Models beyond a Fixed-Length Context." *ACL (1)*. 2019.
  - [4] Yang, Zhilin, et al. "Xlnet: Generalized autoregressive pretraining for language understanding." *Advances in neural information processing systems* 32 (2019).
  - [5] Fan, Angela, et al. "Addressing some limitations of transformers with feedback memory." *arXiv preprint arXiv:2002.09402* (2020).

### Originality:

- To my knowledge, no one has applied self-supervision for this particular domain.

### Quality:

- Overall, it is a good paper. However, there are some issues I mentioned as a Minor.

### Clarity:

- Overall, the paper is well written. However, the location of tables and figures in the document is quite distant from the text.

### Significance:

- I do not find this work a significant contribution due to the lack of new insight or phenomena on how neural networks work compared to related literature (e.g., [6]).
  - Nguyen, Thao, Maithra Raghu, and Simon Kornblith. "Do Wide and Deep Networks Learn the Same Things? Uncovering How Neural Network Representations Vary with Width and Depth." International Conference on Learning Representations. 2021.

**Time Spent Reviewing:**

8

---

> ### Author Response · Authors · 2021-08-06
> **Author Response to Reviewer VbuX**
>
> Thank you for the detailed review. We appreciate that you highlight the quality of the paper as well as the extensiveness of the experiments. Further, we agree that this type of work was not done before and therefore is novel for the research community.
> In the following, we would like to address the points you raised and try to resolve them. We’ll begin with the major points and hope to add clarification. We respond to the minor points afterwards.
>
>
> ## Major
>
> ### Motivation and Significance
> In the paper, we motivate our approach in lines 46 and following. We will try to improve it further and make it sharper. We also discuss this topic in our response to reviewer Te8F, the responses overlap in parts.
> As you confirm, the proposed approach is novel since to the best of our knowledge there exists no other work on representation learning of NN weight spaces. In this context, we would refer to reviewer 6PDp’s description of this submission as “a discovery kind of paper”.
> With our work, we aim to investigate how populations of models behave in weight space. The underlying assumption of our work is that populations of neural networks develop structures (e.g. a manifold) in weight space during training. These structures contain meaningful properties of the models embedded there. With the proposed work, we demonstrate that we can learn these structures efficiently and show by linear probing that we can recover properties of individual models of the population. This adds an new perspective to the existing pairwise comparison of e.g., Nguyen et al., 2021. To us, this is significant, and we’d be more than happy to discuss this, what implication it has, and how we could utilize it further.
>
> Potential applications of this work might be model analysis and monitoring, prediction of training progress, or to generate weights for model initialization. If we understand your point about different evaluations correctly, it indeed goes in a similar direction. One potential application is to exploit the representations for the prediction of model performance at later epochs. Other ideas are to use the learned representation as a regularization in between backpropagation steps.
>
> ### Self-Supervised Learning Approach
> We motivate the use of Self-Supervised Learning in lines 56 and following, but will make this clearer, too. Our goal is to learn task-agnostic, generic representations, that contain rich and diverse information and are exploitable for multiple downstream tasks.
> The direct supervised learning approach has been demonstrated by Unterthiner et al, 2020 and Eilertsen et al., 2020. Unterthiner et al. find that statistics of the weights (mean, var and quintiles) are the best features (among those explored) to predict test accuracy. Our own investigation found a very strong correlation of those features with this one downstream task. However, we intended to learn representations of the weights, that contain information beyond simple statistics.
> As a generalization, ‘labels’ can be obtained relatively simply or exist already. However, they can only describe predefined characteristics of a model instance (e.g., accuracy). Applying multi-task learning may mitigate this, but only if the labels don’t correlate. However, self-supervised approaches allow to learn more information-rich representations (e.g., LeCun and Misra; Self-supervised learning: The dark matter of intelligence; 2021). As we demonstrate, these representations are very useful to predict multiple model characteristics.
>
> ## Minor
> We’ll respond to the points under ‘minor’ in the order you’ve posted them.
> -	Using different seeds in the way of an augmentation is an interesting idea, which we’ll give thought to.  Our current understanding is that seeds define one of the generating factors of the model zoos (among others), and augmentations should be chosen without the knowledge of these generating factors. Additionally, we are of the opinion that augmentations should define means to quickly generate new instances of the original sample with identical (permutation) or similar (noise, erasing) properties.
> -	Thank you for pointing out the learning procedure. We will simplify and clarify that aspect.
> -	Whether the E_c+ setup can be considered contrastive is an interesting question. We tend to yes, as other methods without explicit negative samples (e.g., BYOL or SiamSiam) are often considered contrastive learning. Our line of development was a simplification of SimCLR, as we have stated it. We’ll add references to SiamSiam and SPR.
> -	Regarding model selection, we will change the wording to make it less confusing. Models are selected based on the train split performance only. To better understand generalization, we report the test performance.
> -	Thanks for pointing out the transformer section. We mean a global field of visibility in each transformer layer but are aware of issues with long-range relations. We will clarify that section.
>
> We hope we could add clarification and explanation to the points your raised, and look forward to discussing our work further with you.

---

> > ### Comment · Reviewer_VBuX · 2021-08-15
> > **[–] Rebuttal Response**
> >
> > Thank you for clarifying most of my questions, and I agree with your points on self-supervision. The manuscript has a good amount of work. But I think the only last concern is that the paper still does not look exciting yet in terms of giving insights on how NNs work or exploring the real case scenario (e.g., the one I have proposed).
> >
> > I am willing to increase the score to Marginally above the acceptance threshold (6).

---

> > > ### Author Response · Authors · 2021-08-18
> > > **Thanks for your response**
> > >
> > > Thank you for the response and for changing your score. We’re glad we could clarify most of your questions. If there are remaining open questions, we’d be happy to try to answer then.

---

### Official Review · Reviewer_6PDp · 2021-07-15

**Rating:** 6
**Confidence:** 4

**Summary:**

This is a very interesting paper. I need to be sure that I understand the main concept and review again carefully.

My review rating is really a placeholder, please **ignore my review rating**. I will give a true rating after the authors response to my queries.

Let me know if my understanding of this work is accurate. In a nutshell:

1. Train a zoo of M Neural Net, each with weights w_i, i=1,. . M.
2. All of the w_i has the same dimension N
3. Consider one of w_i as one 'data point' and devise auto encoder kind of network to extract the latent space of w_i
4. let latent space of w_i be z_i, z_i=g_\theta(w_i)
5. analyse these z_i using a linear regressor on downstream tasks -- line 174 in appendix
6. downstream tasks are accuracy prediction, generalisation gap etc

essentially if the latent variables z_i of the weights correlates well with (line 174 in appendix) downstream tasks, e.g. good R^2, we can say that the NN zoo (statistically) picks up good information about the tasks at hand.


**Ethical Concerns:**

NIL

**Main Review:**

I see this paper as more of a discovery kind of paper than a paper for an immediate real world application. However I wish to ask if the author has some eventual real world applications in mind that inspire this work.

The motivation of this work was discussed in lines 46-63. Would be good to strengthen this part of the paper further.

**Time Spent Reviewing:**

3 hours

---

> ### Author Response · Authors · 2021-08-06
> **Author Response to Reviewer 6PDp**
>
> Thanks a lot for your questions, we appreciate your interest in the proposed work.
>
> ### Approach
> Your understanding of the work and break-down exactly represents the contribution of our paper.
> For the sake of completeness, we hope you don’t mind if we add a few details:
> -	Step 3: the representation learning part isn’t limited to autoencoders, although we settled on architectures and losses that at least include reconstruction as one part.
> -	Step 5: to evaluate how well properties are encoded in the learned representations, we apply either a linear regressor or classifier, depending on the property (comp. lines 174 and 218 of the appendix).
> -	Step 6: The trained and frozen encoders and probes are applied to predict properties, i.e., accuracy, epoch, hyperparameters and generalization gap, on the test splits of the same zoo (in-distribution) and without fine-tuning to other zoos (out-of-distribution).
>
> As you said, a high R^2 score shows that the characteristics are encoded linearly in the learned representation. To us this indicates that populations of NN models develop a structure in weight space. This structure can be learned and contains meaningful information on the characteristics of the models embedded in it.
>
> ### Main Review
> We agree, this line of work is exploratory and has not been done before. The proposed approach can be the first step towards a future real-world applications, as such structures can be exploited in (i) the analysis of models, e.g. as monitoring services during training, which score overall performance or predict look-ahead performance at future epochs; (ii) in the analysis of sequences of models as trajectories in representation space, e.g. for versioning, to measure similarities between models; and (iii) sampling from the learned representation to generate models with specific properties, i.e. as an initialization scheme.
>
> We will try to strengthen the motivation in the introduction further.
>
> We hope this addresses the first questions you had to your satisfaction and look forward to a good discussion.

---

> > ### Comment · Reviewer_6PDp · 2021-08-26
> > **further exploration**
> >
> > thank you for the replies. I understand this line of work can go on forever and this is the first step.
> >
> > I encourage the authors to explore a lot more, in their future publications.
> > 1. how does the empirically 'optimised' weights change with respect to number of data points. is there a fixed point and what is the convergence to the fixed point as number of data point increases
> > 2. would be very impactful if there can be some studies and conclusion about how the ML task and data set affects the optimised weight space.
> > 3. is there a clustering of tasks and data set? - we know that two data set that differs infinitesimally should have infinitesimal difference in the optimised weight space

---

> > > ### Author Response · Authors · 2021-08-27
> > > **Thanks for your response**
> > >
> > > Thanks for the reply. We appreciate the expressed interest and the understanding that this work is the first step towards many possible directions that can evolve to additional extensions relevant across a broad set of domains. We share our belief that this research topic may turn to be very fruitful, and we agree that we are at the very beginning of the exploration.
> > >
> > > We also thank you for the recommendations, these are interesting questions to evaluate. Similar to these we have some ideas that are not far from the suggested ones, e.g., work towards the detection of biased (or poisoned) data set. We are also exploring experimental setups to predict development of models to save compute as mentioned by Reviewer VBuX.

---

> > > > ### Comment · Reviewer_6PDp · 2021-09-10
> > > > **acknowledge the response**
> > > >
> > > > I like to thank the authors for putting in a lot of hard work to produce this wonderful piece of work. also thank for authors for carefully response to my reviews.

---

### Official Review · Reviewer_Te8F · 2021-07-16

**Rating:** 6
**Confidence:** 3

**Summary:**

The authors propose three data augmentations and one attention module to learn a representation of the network weights to predict the network accuracy.

**Limitations And Societal Impact:**

The authors described the limitations of the work. And as they mentioned in the checklist, potential negative societal impact does not apply to a single method.

**Main Review:**

Originality: The authors propose three data augmentations. The only novel one seems to be the permutation augmentation. The other two (erasing, noising) seems to be proposed for general self-supervised learning before. The attention modules is also not new. But I think applying self-supervised learning to neural network accuracy prediction itself is new.

Quality:
1. The motivation is still not very clear. If predicting neural network accuracy from the weights can help understand fundamental characteristics that make NN successful, how does using a self-supervised representation help more on the understanding of the fundamental characteristics. Maybe the authors can give some examples on which understanding the authors gain more on the fundamental characteristics.
2. I am not sure why we need to use self-supervised representations. I guess for a different random seed, we may get a different model. And so, the data is unlimited.
3. The experiment results seems to be okay. Self-supervised representation seems to be get better performance on predicting the network accuracy.
4. The equation under line 96 does not look correct. I do not think matrix multiplication and sigmoid function is exchangeable. Also, I am not very sure about why we want this permutation augmentation.(but this does not affect my score)

clarity:
1. The paper is clearly written.

Significance:
1. It's not clear to me "predicting neural network accuracy from the weights can help understand fundamental characteristics that make NN successful". If the authors can give some examples, it will make the contribution more significant.



**Time Spent Reviewing:**

4

---

> ### Author Response · Authors · 2021-08-06
> **Author Response to Reviewer Te8F**
>
> Thank you for your review. Below, we’ll try to address the points you raise in the hope we can clarify some of them. We’ll begin with the motivation and significance, continue with our reasoning to use Self-Supervised Learning and at the end explain why we think the permutation augmentation is both useful and valid.
>
> ### Motivation and Significance
> In the submission, we motivate our approach in lines 46 and following. We will try to improve it further to make it sharper.
> We consider our approach novel since to the best of our knowledge there is no other work on representation learning of NN weigh spaces (as confirmed by reviewer VbuX). In this context, we would refer to reviewer 6PDp’s description of this work as “a discovery kind of paper”.
> The underlying assumption of our work is that populations of neural networks develop structures (e.g., a manifold) in weight space during training. These structures contain meaningful properties of the models embedded there. With the proposed work, we demonstrate that we can learn these structures efficiently and show by linear probing that we can recover properties of individual models of the population.
> To us, this is significant, and we’d be more than happy to discuss it, what implication it has, and how we could utilize it further.
>
> As exploratory work, this is only the first step and we look forward to potential applications in areas such as model analysis and monitoring, insights in learning dynamics and sampling and generation of models. More about this point can be found in our responses to reviewers 6PDp and VbuX.
>
> ### Self-Supervised Learning Approach
> We motivate the use of Self-Supervised Learning in lines 56 and following, but will make this clearer, too. Our goal is to learn task-agnostic, generic representations, that contain rich and diverse information and are exploitable for multiple downstream tasks.
> The direct supervised learning approach has been demonstrated by Unterthiner et al, 2020 and Eilertsen et al., 2020. Unterthiner et al. find that statistics of the weights (mean, var and quintiles) are the best features (among those explored) to predict test accuracy. Our own investigation found a very strong correlation of those features with this one downstream task. However, we intended to learn representations of the weights, that contain information beyond simple statistics.
> As a generalization, ‘labels’ can be obtained relatively simply or exist already. However, they can only describe predefined characteristics of a model instance (e.g., accuracy), whereas self-supervised approaches allow to learn information-rich representations (e.g., LeCun and Misra; Self-supervised learning: The dark matter of intelligence; 2021). As we demonstrate, these representations are very useful to predict model characteristics.
>
>
> ### Permutation Augmentation
> To answer your questions about the permutation augmentation, we would like to split it into two parts. We first address the relevance, then the mathematical validity.
>
> While new data (models) can be generated (trained), it’s computationally expensive. Hence the permutation augmentation. It allows us to (i) cheaply create valid samples, which we empirically find very useful to learn generalizing representations (comp. Table 3); and (ii) add inductive biases via contrastive learning to learn representations, that are invariant to the symmetries in weight space and show higher downstream task performance (comp. Table 10). As mentioned in line 100 and following, an analogy for the permutation augmentation are flips or rotations of images, to which we generally desire model predictions to be invariant, as it’s essentially the same sample.
>
> As for the equation under line 96, we would kindly refer you to the first section of the appendix, where we provide a proof and empirical evaluation of the permutation equivalence for both forward and backward pass. If we have left something unclear, please let us know.
> The operation of drawing the permutation matrix in the activation function does not work for any matrix but is valid for the specific properties of permutation matrices. The activation function is applied element-wise on each output of a neuron. A permutation changes the row-order of the output vector, but not the value of the elements or their computation. As changing the order before or after applying the activation function does not change the result, permuting the argument instead of the output is valid.
>
> We hope to have addressed the points you raise in your review to your satisfaction, added clarification and explanation and look forward to discussing our work further with you.

---

> > ### Comment · Reviewer_Te8F · 2021-08-16
> > **Thanks for the clarifications.**
> >
> > Thanks for the authors' response. Some of my questions are addressed. I agree that the equation under line 96 is correct. But I still do not quite get the significance of the work. As the authors mentioned, this work is a  “a discovery kind of paper”. But I am not sure what the readers can gain more over the original paper that proposed we can predict model characteristic prediction from network weights. I totally agree with Reviewer VBuX "the paper still does not look exciting yet in terms of giving insights on how NNs work or exploring the real case scenario ". So I still think the paper is on the borderline and am not going to change my score.

---

> > > ### Author Response · Authors · 2021-08-18
> > > **Thanks for your response**
> > >
> > > Thank you for the response. We’re glad we could clarify some of your questions. If other questions are still open, as your response implies, let us know, we’re looking forward to answering them.
> > >
> > > We appreciate your questions towards providing additional gains and real case scenarios of the presented approach. As one immediate example, at this year’s ICML, the following two papers have been published that investigate subspaces within the weight space of neural networks:
> > > - “On Monotonic Linear Interpolation of Neural Network Parameters”; James Lucas, Juhan Bae, Michael Zhang, Stanislav Fort, Richard Zemel, Roger Grosse
> > > - “Learning Neural Network Subspaces”; Mitchell Wortsman, Maxwell Horton, Carlos Guestrin, Ali Farhadi, Mohammad Rastegari
> > >
> > > Both papers identify linear subspaces of the weight space with useful properties. As we motivate our work similarly, but in contrast learn nonlinear representations of the weight space, our work appears directly connected and we think that it is beneficial for the community.

---

### Official Review · Reviewer_Rn7M · 2021-07-16

**Rating:** 7
**Confidence:** 3

**Summary:**

This paper proposes a self-supervised representation learning approach to uncover properties from trained neural network (NN) weights and biases. Representations are learnt through encoder-decoder architectures on the weights, using various losses. Attention-based modules are used to find interactional structure between different weights of a NN, resulting in compressed representations in different manners. To this end, data augmentation based on permutation augmentation is proposed; random erasing and noise augmentation are also explored. Empirics are performed on trained models from MNIST, Fashion-MNIST, SVHN, CIFAR-10 as well as synthetic datasets. Results explore data augmentation, architecture ablation, out-of-distribution prediction and various downstream tasks.

**Limitations And Societal Impact:**

The limitations and societal impacts are discussed.

**Main Review:**

Overall, the paper is well written and clear. The topic is relevant to the literature and its goal relevant. The methodology is well described and seems sound. The experimental setting is detailed and experiments are extensive. The addition of a few details would improve the quality of the paper:
- the amount of compute needed is mentioned. Could you also report model sizes?
- references to the experiments developed in the appendix could be added (eg. on the impact of the compression ratio)

I recommend an accept for this paper.

Typos:
- line 57: ", so" instead of ". So"
- line 114: "a self-supervised fashion"
- citing types are often mixed up (eg lines 113, 120)


**Time Spent Reviewing:**

2

---

> ### Author Response · Authors · 2021-08-06
> **Author Response to Reviewer Rn7M**
>
> Thank you very much for your review. We appreciate the acknowledgment of our contribution.
>
> As for the points you raised:
> -	We will compute the parameters in the models we used in the days to come, report them here and add them to the paper. Additionally, we will publish the zoos and learned representations on github.
> -	Further references to the experiments in the appendix will be added, thank you for pointing that out.
> -	Thanks for bringing the typos to our attention. They will be fixed.
>
> If there is anything else we can clarify, we’re glad to do so and look forward to your message.

---

> > ### Author Response · Authors · 2021-08-10
> > **Representation Learning Model Sizes**
> >
> > As promised earlier, we here provide a table with the number of parameters of the representation learning models used across our experiments. We used the same representation learning model architecture on all model zoos by Unterthiner et. al (MNIST-HYP, FASHION-HYP, SVHN-HYP, CIFAR-HYP).
> > While our focus in this work was to show that representation learning for this domain has merits, we also see potential future work in a direction of reducing the size of the representation learning models.
> >
> >
> > | Zoo | # of parameters |
> > | --- | --- |
> > | TETRIS-SEED  | 225'910 |
> > | TETRIS-HYP  |  6'606'407 |
> > | MNIST-SEED  | 63'365'060 |
> > | MNIST-HYP-1-FIX-SEED   | 60'692'566 |
> > | MNIST-HYP-1-RAND-SEED   | 60'122'610 |
> > | MNIST-HYP-5-FIX-SEED   | 60'122'610 |
> > | MNIST-HYP-5-RAND-SEED  | 60'122'610 |
> > | Zoos by Unterthiner et al. | 18'983'149 |

---

### Decision · Program_Chairs · 2021-09-27

**Decision:**

Accept (Poster)

**Comment:**

There was a robust discussion between the reviewers and the authors and some discussion amongst the reviewers themselves. The post-rebuttal consensus is that we should accept this work. From the discussion with reviewers, some of the constructive feedback for a final version would be to show more evidence of usefulness in real case scenarios. It would also be useful to explore the directions suggested by reviewer 6PDp in their conversation with the authors.